# Molecular analysis and essentiality of Aro1 shikimate biosynthesis multi-enzyme in *Candida albicans*

Peter J Stogios[1],*, Sean D Liston[2],*, Cameron Semper[3],*, Bradley Quade[4], Karolina Michalska[5], Elena Evdokimova[1], Shane Ram[3], Zbyszek Otwinowski[4], Dominika Borek[4], Leah E Cowen[2], Alexei Savchenko[1,3]

In the human fungal pathogen *Candida albicans, ARO1* encodes an essential multi-enzyme that catalyses consecutive steps in the shikimate pathway for biosynthesis of chorismate, a precursor to folate and the aromatic amino acids. We obtained the first molecular image of *C. albicans* Aro1 that reveals the architecture of all five enzymatic domains and their arrangement in the context of the full-length protein. Aro1 forms a flexible dimer allowing relative autonomy of enzymatic function of the individual domains. Our activity and in cellulo data suggest that only four of Aro1's enzymatic domains are functional and essential for viability of *C. albicans*, whereas the 3-dehydroquinate dehydratase (DHQase) domain is inactive because of active site substitutions. We further demonstrate that in *C. albicans*, the type II DHQase Dqd1 can compensate for the inactive DHQase domain of Aro1, suggesting an unrecognized essential role for this enzyme in shikimate biosynthesis. In contrast, in *Candida glabrata* and *Candida parapsilosis*, which do not encode a Dqd1 homolog, Aro1 DHQase domains are enzymatically active, highlighting diversity across *Candida* species.

## Introduction

Fungal, archaeal, bacterial, plant, and apicomplexan species have the capacity for de novo synthesis of aromatic amino acids from carbohydrate precursors. The linear portion of this biosynthetic cascade is known as the shikimate pathway. It combines seven consecutive enzymatic reactions to convert phosphoenolpyruvate and erythrose-4-phosphate into chorismate (Fig 1A). Chorismate is the branch point for synthesis of aromatic acids (L-tryptophan, L-phenylalanine, and L-tyrosine) and their derivatives, including vitamin K, ubiquinone, and *p*-aminobenzoate.

Shikimate biosynthesis has been the focus of intense studies because of its essential role in microbial and plant cell metabolism and it serves as a segue pathway for the synthesis of biotechnologically important aromatic compounds (Gosset, 2009; Liu et al, 2020). This pathway is absent in metazoa, including humans, which instead require exogenous aromatic amino acids as part of their diet. This makes the shikimate pathway an attractive target for development of herbicides and antimicrobial agents (Coracini & de Azevedo, 2014; Duke, 2018; Nunes et al, 2020). Glyphosate, which inhibits the plant 5-enolpyruvylshikimate-3-phosphate synthase (EPSPS) that catalyses the sixth step in the shikimate pathway (Fig 1A), has been developed into one of the most successful agricultural herbicides widely used for weed management (Duke, 2018). Along the same lines, active programs have been mounted to identify and optimise inhibitors of the bacterial shikimate pathway enzymes (Coggins et al, 2003; Cheng et al, 2012; Blanco et al, 2013; Gordon et al, 2015; Sahu et al, 2020). However, antimicrobials based on shikimate pathway inhibition are yet to be introduced to the clinic.

Recognising their potential as antimicrobial targets, the shikimate biosynthesis enzymes have been characterised in multiple microbial species, revealing several conserved features (Mir et al, 2015). For example, 3-deoxy-D-arabinoheptulosonate-7-phosphate (DAHP) synthase, which catalyses the first step of the pathway, is represented by multiple isoenzymes that are subject to feedback inhibition by individual aromatic amino acid products. The Gram-negative bacterium *Escherichia coli* encodes three DAHP synthases that are feedback inhibited by L-phenylalanine, L-tyrosine, and L-tryptophan, respectively (Staub & Denes, 1969). The model yeast *Saccharomyces cerevisiae* and pathogenic fungus *Candida albicans* produce two DAHP synthases, which are inhibited by L-phenylalanine and L-tyrosine, respectively (Teshiba et al, 1986; Pereira & Livi, 1993; Sousa et al, 2002; Konig et al, 2004). Despite similarity in primary sequence, there is significant diversity in the organisation of enzymatic domains among shikimate pathway enabled species. In Fungi and Apicomplexa, the second to the sixth steps of the shikimate pathway are supported by five enzymatic domains combined in a single protein known as Aro1 or AroM (Graham et al,

[1]Department of Chemical Engineering and Applied Chemistry, University of Toronto, Toronto, Canada   [2]Department of Molecular Genetics, University of Toronto, Toronto, Canada   [3]Department of Microbiology, Immunology and Infectious Diseases, University of Calgary, Calgary, Canada   [4]Department of Biophysics, University of Texas Southwestern Medical Center, Dallas, TX, USA   [5]Structural Biology Center, X-ray Science Division, Argonne National Laboratory, Argonne, IL, USA

Correspondence: alexei.savchenko@ucalgary.ca
*Peter J Stogios, Sean D Liston, and Cameron Semper contributed equally to this work.

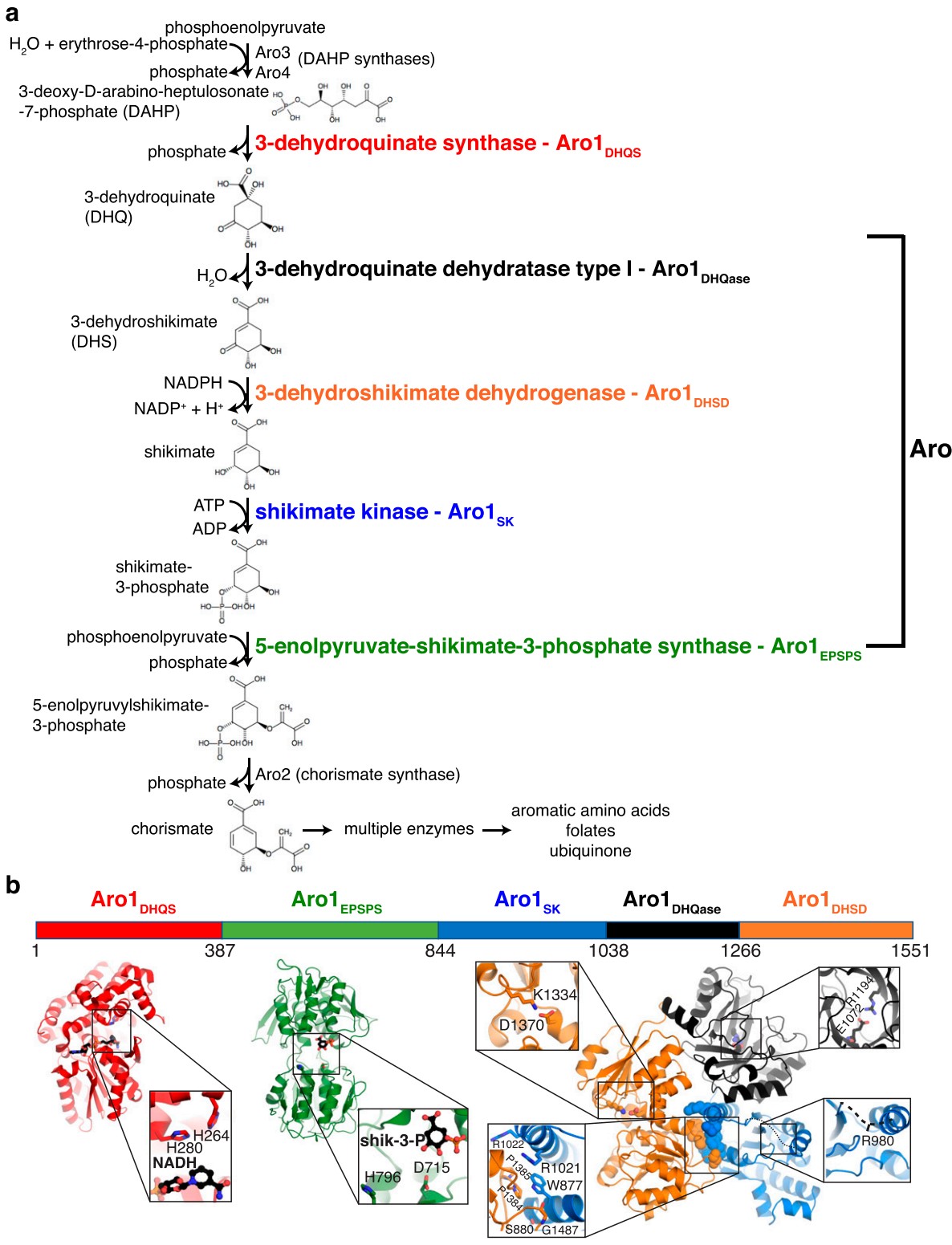

**Figure 1. *C. albicans* shikimate pathway.**
**(A)** Shikimate pathway, with enzyme names indicated to the right of each reaction. Aro1 individual domains are highlighted in colours. **(B)** Crystal structures of Aro1 fragments solved in this work, including Aro1_DHQS, Aro1_EPSPS, Aro1_SK+DHQase+DHSD. Zooms show active sites, key catalytic residues, or domain–domain interactions in the Aro1_SK+DHQase+DHSD structure.

1993; Campbell et al, 2004). In plants and bacteria these activities are typically performed by individual single-domain enzymes: 3-dehydroquinate synthase (DHQS), 3-dehydroquinate dehydratase type I (DHQase), 3-dehydroshikimate dehydrogenase (DHSD), shikimate kinase (SK), and 5-enolpyruvate-shikimate-3-phosphate synthase (EPSPS). DHQS (Liu et al, 2008; Neetu et al, 2020) and type I DHQase (Nichols et al, 2004b; Light et al, 2011).

The biological consequence(s) of encoding fungal and apicomplexan shikimate biosynthesis enzymes as a multi-domain protein remain poorly understood. AroM from the thermophilic soil-dwelling fungus *Chaetomium thermophilum* is the only fully structurally characterised representative of a penta-functional shikimate biosynthesis protein (Verasztó et al, 2020). This multi-enzyme forms a stable dimer with two of the domains (DHQS and DHQase/DHQD) mediating inter-subunit interactions, reminiscent of the interactions described for the corresponding single-domain shikimate pathway enzymes in bacteria. Surprisingly, *C. thermophilum* AroM showed no evidence for increased metabolic throughput in vitro when compared with isolated enzymatic domains (Verasztó et al, 2020). However, significant sequence variation exists between AroM/Aro1 homologs—specifically, 52% identity between *C. thermophilum* AroM and *C. albicans* Aro1—and limited enzymatic characterization necessitates further structural and functional study of multi-domain complexes involved in chorismate biosynthesis, particularly in clinically important species such as *C. albicans* (Verasztó et al, 2020).

*C. albicans* is a member of healthy human microbiota, but can also cause life-threatening systemic fungal infections, which occur predominantly in immunocompromised people (Low & Rotstein, 2011; Bongomin et al, 2017). Mortality from these infections remains alarmingly high at ~40% despite current state-of-the-art treatments (Pfaller & Diekema, 2010). The limited selection of antifungal therapies is further threatened by the evolution of drug resistance in *C. albicans* (Lee et al, 2021), and emergence of other drug-resistant *Candida* species, such as *Candida auris* (Lockhart et al, 2017). To address these challenges, it is essential to identify targets to exploit in novel antifungal therapies.

Hundreds of genes essential for viability, morphogenesis, and host cell escape have been identified in *C. albicans* using genomic resources, including the *C. albicans* gene replacement and conditional expression (GRACE) library that covers close to half of the protein coding genes (Roemer et al, 2003; O'Meara et al, 2015; O'Meara et al, 2018; Fu et al, 2021). "Fungal-specific" genes with no identifiable mammalian homolog that encode proteins essential for growth and/or pathogenesis are of particular interest as targets for antifungal drug development (Roemer et al, 2003; Becker et al, 2010; O'Meara et al, 2015; Segal et al, 2018). These studies identified the genes *ARO1*, *ARO2*, and *ARO7* encoding for non-redundant enzymes in the shikimate pathway as essential *in cellulo* and in a murine model of disseminated candidiasis (Roemer et al, 2003; Becker et al, 2010; Fu et al, 2021). A recent study demonstrated that transcriptional repression of *ARO1* in *C. albicans* results in a complex phenotype including changes to cell wall integrity and biofilm formation, and attenuated virulence of *C. albicans* in the *Galleria mellonella* infection model (Yeh et al, 2020). Furthermore, *ARO1* was classified as essential based on mapping transposon insertion sites in a stable haploid isolate of *C. albicans* (Segal et al,

2018), and confirmed in this study (Fig S1). These observations identify the Aro1 multi-enzyme as essential for *C. albicans* viability and pathogenesis; however, the role of individual enzymatic domains in this protein remained unknown until now.

Characterization of the five enzymatic activities of Aro1 and their essentiality for fungal growth is critical to define target sites for small molecule inhibitors with potential therapeutic relevance. Here, we present the molecular structure of all five Aro1 domains across crystal structures of three Aro1 fragments and a cryo-EM–derived model of the full-length protein. These structures reveal a homodimer mediated by the DHQase domain and highlight extensive interdomain flexibility. Surprisingly, we identified that the *C. albicans* Aro1 (type I) DHQase domain harbors sequence substitutions in its catalytic site that render it inactive. We suggest that this step in the pathway may be supported instead by the type II DHQase Dqd1 enzyme in *C. albicans*, which showed robust activity against shikimate pathway intermediates in our in vitro assays. Finally, leveraging a conditional expression system, we show that all four functional domains of Aro1 are individually essential for growth of *C. albicans*. Therefore, we propose that each of these four enzyme domains of *C. albicans* Aro1 may be targeted by small molecule inhibitors for development of novel antifungal therapies.

# Results

## Crystal structures reveal active site architectures of all five domains of *C. albicans* Aro1

To provide molecular details into the essentiality of *C. albicans* Aro1, we first pursued its structural characterization. For this we designed expression constructs for full-length *C. albicans* Aro1 and fragments corresponding to individual enzymatic domains N-terminally fused to polyhistidine-TEV protease cleavage site (His$_6$TEV) tag in *E. coli*. The expression construct for full-length *C. albicans* Aro1 was unstable in *E. coli*, likely due to toxicity that was abrogated through a H268K/H280K double mutation. We were unable to express the wild-type full-length Aro1 protein but we were able to express and purify three Aro1 fragments that together encompass all five enzymatic domains. These fragments included the N-terminal 387 residues corresponding to the DHQS domain (Aro1$_{DHQS}$); residues 387–844, corresponding to the EPSPS domain (Aro1$_{EPSPS}$); and the C-terminal residues 845–1,551, comprising the SK, DHQase, and DHSD domains (Aro1$_{SK-DHQase-DHSD}$) (Fig 1). These three fragments were submitted to crystallisation and crystal structures of Aro1$_{DHQS}$, Aro1$_{EPSPS}$, and Aro1$_{SK-DHQase-DHSD}$ were determined to 1.85, 1.85, and 2.3 Å, respectively, by molecular replacement (x-ray crystallographic statistics in Table S1).

Aro1$_{DHQS}$ was crystallised in the presence of its cofactor NADH. Accordingly, in addition to the two copies of this domain found in the asymmetric unit we identified unambiguous additional electron density corresponding to one NADH molecule bound to each copy of this fragment (Figs 1B and S2). The Aro1$_{DHQS}$ structure was most similar to the corresponding domains in previously characterised AroM proteins from *Aspergillus nidulans* (Carpenter et al, 1998; Nichols et al, 2003, 2004a) and from *C. thermophilum* (PDB 6HQV

[Verasztó et al, 2020]) (Fig S2). Aro1$_{DHQS}$ also shared significant structural similarity with bacterial DHQS enzymes from *Acinetobacter baumannii* (PDB 5EKS), *Vibrio cholerae* (PDB 3OKF), *Mycobacterium tuberculosis* (PDB 3QBE), and *Staphylococcus aureus* (PDB 1XAL; [Nichols et al, 2004c]). Comparison of these structures with the Aro1$_{DHQS}$-NADH complex suggests that the latter follows the "open" binding mode characteristic of DAHP-free structures of DHQS (Fig S2). Notably, the residues that interact with the 3-dehydroquinate analog carbaphosphonate, NADH, and Zn$^{2+}$ as observed in the *A. nidulans* AroM$_{DHQS}$ structure are completely conserved in the Aro1$_{DHQS}$ active site (Fig 2A). Specifically, Aro1$_{DHQS}$ residues H264, H268, and H280 coordinate the catalytic Zn$^{2+}$, allowing for identification of potential catalytic residues of Aro1. As mentioned above, the two Aro1$_{DHQS}$ chains in the asymmetric unit formed a dimer with over 1,100 Å$^2$ buried surface area (Fig S2). The

two corresponding domains in the fungal AroM multi-enzymes showed very similar dimeric arrangements (Fig S2), along with the single-domain DHQS enzymes listed above.

For the Aro1$_{EPSPS}$ fragment structure, we identified two copies of this domain in the asymmetric unit and they adopt a near identical conformation. The regions corresponding to residues 784–794 and 819–823 were disordered and could not be modeled. The structure of Aro1$_{EPSPS}$ (Fig 1B) is composed of two distinct subdomains (I and II) connected by a hinge region. This architecture is consistent with many previously characterised EPSPS enzymes, including those from *C. thermophilum* (PDB 6HQV); (Verasztó et al, 2020) (Fig S2), *V. cholerae* (PDB 3NVS), *E. coli* (1G6T, [Schonbrunn et al, 2001]), and *A. baumannii* (PDB 5BS5). Because the Aro1$_{EPSPS}$ fragment was crystallised in the presence of shikimate-3-phosphate, we identified additional electron density corresponding to a molecule of this

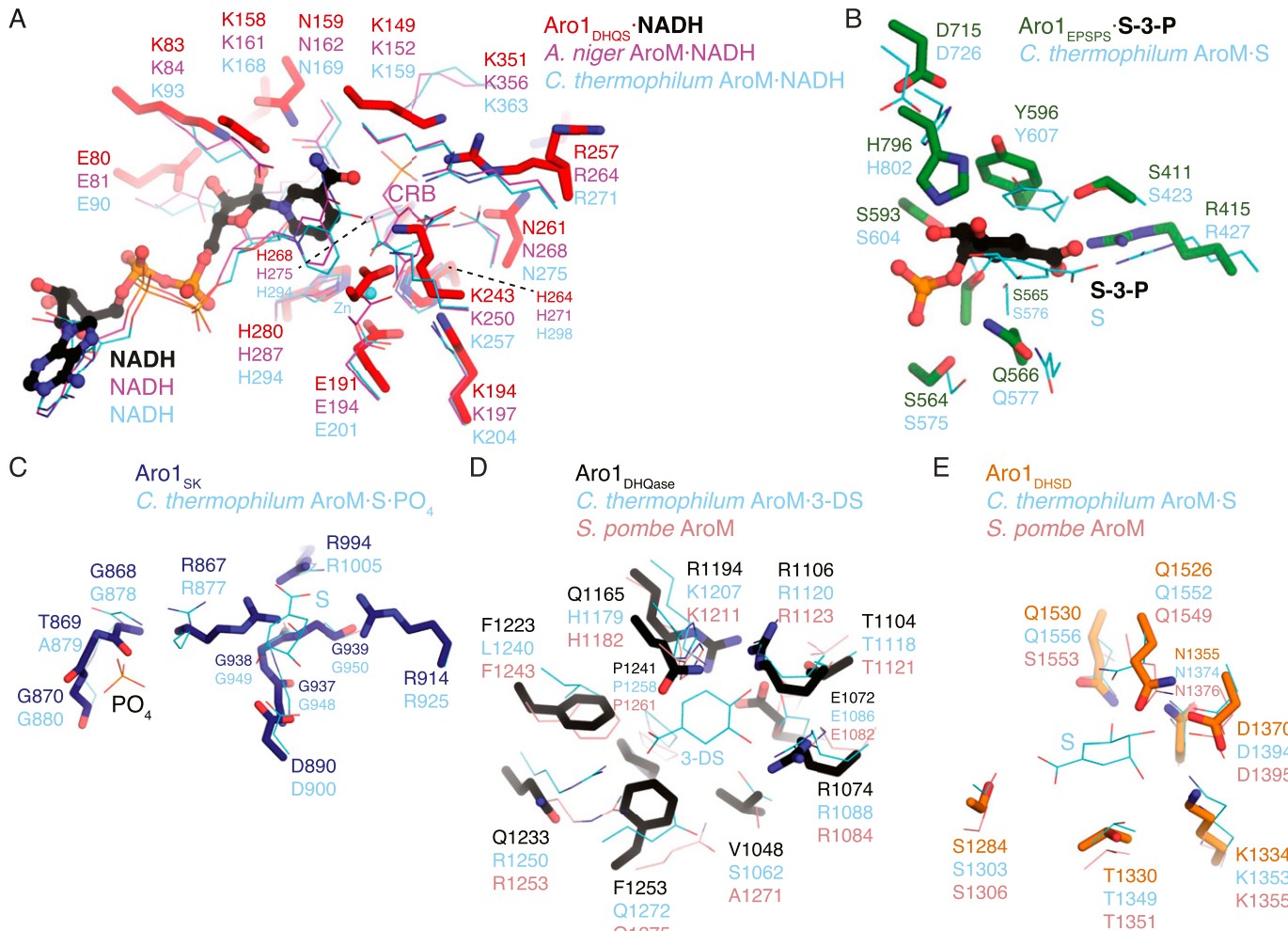

**Figure 2. Molecular architecture of active sites in *C. albicans* Aro1 domains.**
In all panels, Aro1 residues are shown as sticks and the comparative structure's residues are shown as thin lines. **(A)** Aro1$_{DHQS}$ superposed with *A. niger* AroM DHQS domain (PDB 1NVB). NAD molecules bound to the respective structures are shown in sticks. Carbaphosphonate bound to AroM DHQS shown in sticks. Zn2+ ion bound to Aro1DHQS shown as a sphere. **(B)** Aro1$_{EPSPS}$ superposed with *C. thermophilum* AroM EPSPS domain (PDB 6HQV). Shikimate-3-phosphate and shikimate bound to Aro1DHQS and AroM, respectively, shown as sticks. **(C)** Aro1$_{SK}$ from Aro1$_{SK+DHQase+DHSD}$ structure superposed with *C. thermophilum* AroM SK domain (PDB 6HQV). Shikimate bound to AroM shown in sticks. **(D)** Aro1$_{DHQase}$ from Aro1$_{SK+DHQase+DHSD}$ structure superposed with *C. thermophilum* AroM DHQase domain (PDB 6HQV) and *S. pombe* AroM DHQase domain (PDB 5SWV). 3-dehydroshikimate bound to *C. thermophilum* AroM shown in sticks. **(E)** Aro1$_{DHSD}$ from Aro1$_{SK+DHQase+DHSD}$ structure superposed with *C. thermophilum* AroM DHSD (PDB 6HQV) and *S. pombe* AroM DHSD domain (PDB 5SWV).

substrate in the cleft between the two subdomains (Figs 1B and S2). The lobes of EPSPS enzymes are known to transition from open to closed states upon substrate binding and our analysis confirmed that Aro1$_{EPSPS}$-shikimate-3-phosphate complex structure adopted the closed conformation. The interactions with shikimate-3-phosphate in the Aro1$_{EPSPS}$ structure were primarily localised to subdomain II. These molecular interactions identify key catalytic residues of Aro1$_{EPSPS}$ such as D715 and H796. The equivalent residues have been implicated in catalysis of single-domain EPSPS enzymes (Sutton et al, 2016; Lee et al, 2017; Verasztó et al, 2020) as well as in the corresponding domain of *C. thermophilum* AroM (Fig 2B).

The structure of Aro1$_{SK-DHQase-DHSD}$ contained four copies of this fragment in the asymmetric unit. Aro1$_{SK}$ domain spanned residues 857–1,032 and was connected by a short linker (residues 1,033–1,038) to Aro1$_{DHQase}$ (residues 1,039–1,267), which is directly followed by Aro1$_{DHSD}$ (residues 1,268–1,551). Notably, each Aro1$_{DHQase}$ mediated homodimerization within the asymmetric unit and via crystal symmetry (Fig S3).

Aro1$_{SK}$ (Fig 1B) was highly structurally similar to other SK enzymes (Fig S2) (Gu et al, 2002; Hartmann et al, 2006; Sutton et al, 2015). The regions corresponding to residues 844–855 and 970–984 were not modeled because of poor electron density. The latter region corresponds to the "lid loop" or "P-loop" that is involved in positioning of ATP and shikimate in the active site (Gu et al, 2002; Hartmann et al, 2006; Sutton et al, 2015). Based on comparisons with other SK structures, residues 967–984 are expected to coordinate the phosphate groups of ATP and shikimate itself. Also, in accordance with its apoenzyme state, residues 1,011–1,021 were localised away from the active site cleft; these residues are expected to engage with the adenine moiety of ATP. Whereas the disordered regions did not allow for full structural analysis of the Aro1$_{SK}$ active site, all the ATP phosphate and shikimate-binding residues appear conserved with other characterised SK, including this domain in *C. thermophilum* AroM (Fig 2C). Based on this, we hypothesised that D890 in Aro1$_{SK}$ would interact directly with shikimate as observed in bacterial SKs (Gu et al, 2002; Hartmann et al, 2006; Sutton et al, 2015), whereas the R980 residue would be involved in positioning of ATP phosphates (Gu et al, 2002; Hartmann et al, 2006; Sutton et al, 2015). We predicted that both these residues would be critical for Aro1$_{SK}$ activity.

Aro1$_{DHQase}$ domain within the Aro1$_{SK-DHQase-DHSD}$ fragment adopts the TIM barrel fold typical of type I DHQase enzymes and also shows significant similarity with the corresponding domains of the two other structurally characterised fungal AroM proteins (Figs 1B and S2). However, the active site composition of Aro1$_{DHQase}$ shows important differences in comparison with previously characterised type I DHQase enzymes (Fig 2D). Notably, we failed to identify residues that could serve as the catalytic dyad typical of type I DHQase enzymes (Leech et al, 1998; Lee et al, 2002; Light et al, 2013). This catalytic dyad features a histidine that acts as a general acid/base and is conserved in most characterised members of this enzyme family including this domain in AroM proteins (i.e., H1179 and H1182 in *C. thermophilum* and *Schizosaccharomyces pombe* AroM's, respectively, Fig 2D) (Verasztó et al, 2020) (PDB 5SWV). The corresponding position in Aro1$_{DHQase}$ is occupied by a glutamine (Q1165) (Fig 2D). The other residue of the dyad in type I DHQase enzymes is usually represented by a lysine that functions as a nucleophile and forms a covalent bond with 3-dehydroquinate

(Light et al, 2014). With no suitable lysine residue in the Aro1$_{DHQase}$ active site, only the R1194 residue could be implicated for this role. As mentioned, the Aro1$_{SK-DHQase-DHSD}$ fragment structure shows that Aro1$_{DHQase}$ is actively involved in dimerization of this fragment (Fig S3).

Finally, Aro1$_{DHSD}$ within the Aro1$_{SK-DHQase-DHSD}$ fragment shows the two subdomain architecture typical of AroE-type shikimate dehydrogenase enzymes (Figs 1B and S2) (Bagautdinov & Kunishima, 2007; Gan et al, 2007; Hoppner et al, 2013). The N-terminal subdomain adopts an $\alpha/\beta$ fold with a central 6-stranded $\beta$-sheet that usually harbors most of the catalytic apparatus, whereas the C-terminal subdomain adopts the Rossman fold that, in other homologs, provides residues involved in interactions with NAD(P)H. Comparative analysis with equivalent domains of the two fungal AroM enzymes suggested Aro1$_{DHSD}$ residues involved in the shikimate binding (Fig 2E). In particular, Aro1$_{DHSD}$ K1134 and D1370 residues are appropriately positioned to serve as the catalytic dyad identified in other DHSD enzymes, with the lysine acting as a general base and the aspartate stabilizing this residue as well as the 3-dehydroshikimate C4 hydroxyl (Peek & Christendat, 2015).

The crystal structure of Aro1$_{SK-DHQase-DHSD}$ provided the first indication of interdomain interactions in full-length Aro1. According to this structure, each of the three Aro1 domains forms extensive interactions with the other two neighboring domains, specifically involving residues W177, S880, R1021, R1022, P1384, P1385 and G1487 (Fig 1B). The Aro1$_{SK}$ and Aro1$_{DHQase}$ interface covered 1,096 Å$^2$, whereas the Aro1$_{SK}$-Aro1$_{DHSD}$ and Aro1$_{DHQase}$-Aro1$_{DHSD}$ interfaces buried 557 and 542 Å$^2$, respectively. The Aro1$_{SK}$ domain, in particular, made the most intimate contacts with the other two domains including interdigitation of its C-terminal $\alpha$-helix seven via residues R1021 and R1022 into the space between the three domains (Fig 1B). A comparison of this fragment of Aro1 with the structures of full-length *C. thermophilum* AroM and the *S. pombe* AroM DHQase-DHSD fragment revealed significant differences in the relative arrangement of equivalent domains (Fig S4). Specifically, the location of Aro1$_{SK}$ relative to Aro1$_{DHQase}$ appears to be rotated ~30° compared with the arrangement of the corresponding domains in the two AroM protein structures. Along the same lines, the C-terminal lobe of the Aro1$_{DHSD}$ domain also appears rotated by ~30°. This differential positioning, however, could be due to the apo status of the Aro1$_{SK}$ domain, with its $\alpha$-helix 7 lacking interactions with a phosphate group donor molecule allowing it to protrude into the core of the tri-domain structure.

## Analysis of full-length *C. albicans* Aro1 by cryogenic electron microscopy suggests significant interdomain mobility

Although we were unable to express recombinantly wild-type Aro1 in *E. coli*, we succeeded in purifying His$_6$TEV-tagged Aro1 from a *C. albicans* engineered strain described in the next section (see also the Materials and Methods section). The size-exclusion chromatography indicates that Aro1 exists predominantly as a stable dimer under experimental conditions (Fig S5). Analysis of Aro1$_{DHQS}$ and the Aro1$_{SK-DHQase-DHSD}$ fragment X-ray structures presented above (Figs 1B and 2) suggested that Aro1 dimer can be stabilised via its DHQS and DHQase domains and interactions between the SK, DHQase, and DHSD domains. Although we were unable to obtain crystals of *C. albicans* Aro1, we used cryogenic electron microscopy single

particle reconstruction (cryoEM SPR) to investigate the Aro1 structure (Figs 3 and S6 and Table S2). The analysis of this sample under cryogenic conditions revealed a combination of full-length and fragmented particles (Fig 3A), likely due to breakage by the water–air interface during blotting. Nevertheless, we were able to identify a subgroup of particles corresponding to dimers of full-length Aro1. Within these particles, there was further structural heterogeneity, due to the Aro1$_{SK-DHQase-DHSD}$ fragment tilting in various conformations relative to the Aro1$_{DHQS}$ domain. In all classifiable Aro1 particles, including the ones corresponding to Aro1 fragments, we also observed considerable blurring of one of the two Aro1$_{EPSPS}$ domains, indicating extensive mobility of this domain. After 3D classification, we identified 87,484 particles belonging to a stable conformation representing a dimer of full-length Aro1 at resolution of 4.19 Å. We performed local refinement on the dimer of the Aro1$_{SK-DHQase-DHSD}$ region, as well as the dimer of Aro1$_{DHQS}$ and Aro1$_{EPSPS}$ domains that resulted in molecular maps of resolutions of 3.16 Å for Aro1$_{SK-DHQase-DHSD}$ and 3.43 Å for Aro1$_{DHQS-EPSPS}$ (Figs 3B and S6).

The 4.19 Å map that corresponds to the homogeneous refinement based on 3D reconstruction of full-length Aro1 dimer (EMDB: 26357, PDB: 7U5S). In this map, we performed only a rigid body fit of two high-resolution models for Aro1$_{DHQS-EPSPS}$ and Aro1$_{SK-DHQase-DHSD}$ domains. These two models (Aro1$_{DHQS-EPSPS}$ and Aro1$_{SK-DHQase-DHSD}$)

were obtained by docking and refining against cryo-EM maps X-ray crystallographic models described earlier. We performed atomic refinement of X-ray crystallographic models with PHENIX and corrected structures manually with Coot. The molecular maps at lower contour level indicated where the linker between Aro1$_{DHQS-EPSPS}$ and Aro1$_{SK-DHQase-DHSD}$ is located (Fig S6). Analysis of the possible distances for 11 aa unmodeled linker excluded the possibility of domain swapping.

As implied by the crystal structure of isolated Aro1$_{DHQS}$, dimerization of the full-length cryoEM SPR structure of Aro1 is mediated by extensive contacts between the Aro1$_{DHQS}$ domain, resulting in an arrangement similar to the one observed in *C. thermophilum* AroM homodimer. Notably, we did not add any ligands to *C. albicans* Aro1 sample used for our cryo-EM analysis, nor did we observe any evidence of endogenous ligands in the five active sites, in contrast to the crystal structure of *C. thermophilum* Aro1 that contained numerous ligands. The interface between the Aro1$_{DHQS}$ domains in the Aro1 dimer is the same as seen in the Aro1$_{DHQS}$ crystal structure. There are additional contacts between the Aro1$_{DHQase}$ domains in the Aro1 dimer that may also contribute to dimerization. Interestingly, within the Aro1 dimer we observed asymmetry between the monomers, with the Aro1$_{SK-DHQase-DHSD}$ region tilting ~55° relative to the Aro1$_{DHQS}$ and Aro1$_{EPSPS}$ domains (Fig 3C). This large

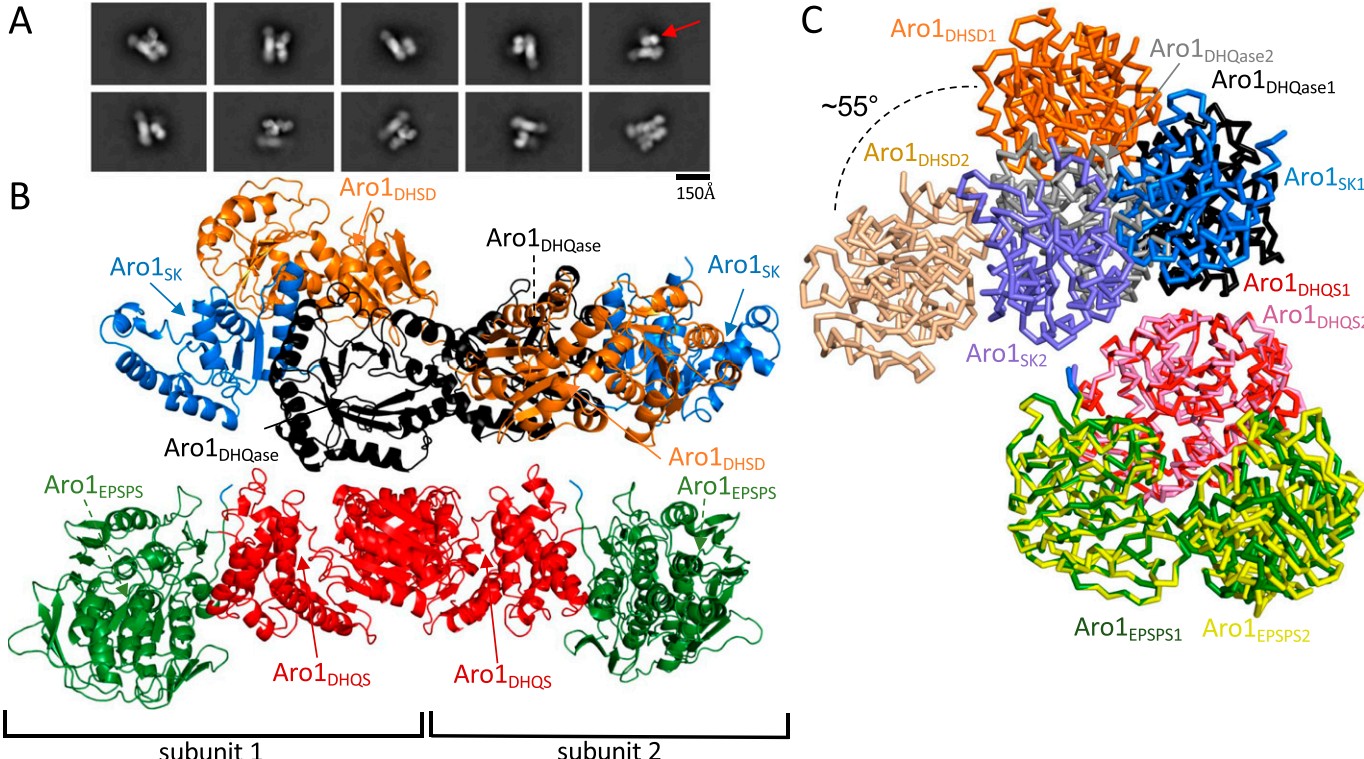

**Figure 3. Cryo-EM structure of the unliganded full-length Aro1 from *C. albicans*.**
**(A)** Selected 2D class averages of the full-length Aro1 dimer. The red arrow points to the Aro1$_{SK-DHQase-DHSD}$ region, which is highly heterogeneous and found tilted at various angles relative to the Aro1$_{DHQS-EPSPS}$ region. **(B)** The structure of the full-length Aro1 from *C. albicans*. The final resolution for the Aro1$_{SK-DHQase-DHSD}$ domain dimer is 3.16 Å and the Aro1$_{DHQS-EPSPS}$ domain dimer is 3.43 Å. Coloured arrows point towards the ligand binding sites of each domain, which are unoccupied in our structure. The Aro1$_{SK-DHQase-DHSD}$ region of subunit 1 is tilted away from the Aro1$_{DHQS-EPSPS}$ region and is referred to as the "loose" monomer, whereas the Aro1$_{SK-DHQase-DHSD}$ region of subunit 2 is nearly parallel to the Aro1$_{DHQS-EPSPS}$ region and is referred to as the "tight" monomer. **(C)** Comparison of "tight" and "loose" monomers from the full-length Aro1 dimer superposed on the Aro1$_{DHQS-EPSPS}$ region. Although the backbones of each domain overlays with low RMSD, the relative orientations of the Aro1$_{SK-DHQase-DHSD}$ and Aro1$_{DHQS-EPSPS}$ regions change dramatically.

rearrangement of the $Aro1_{SK-DHQase-DHSD}$ region results in ~35 Å shift in the $Aro1_{SK}$ domain positioning it directly above subdomain II of the $Aro1_{EPSPS}$ domain. In this conformation the $Aro1_{SK-DHQase-DHSD}$ region in one monomer is nearly parallel to the $Aro1_{DHQS}$ and $Aro1_{EPSPS}$ domains, whereas the other monomer has the $Aro1_{SK-DHQase-DHSD}$ region tilted ~40° relative to the $Aro1_{DHQS}$ and $Aro1_{EPSPS}$ domains.

The analysis of the full-length Aro1 structure highlighted the high level of structural similarity between the specific domains in the Aro1 dimer and their individual crystal structures (Fig S7). The $Aro1_{DHQS}$ domain, $Aro1_{EPSPS}$ subdomain II, and $Aro1_{SK-DHQase-DHSD}$ backbones show nearly identical conformation within the core structure of each region with minor differences seen in some loop regions. This implies that the large movement of the $Aro1_{SK-DHQase-DHSD}$ region does not drive structural changes that modulate substrate binding or catalysis, but instead reorients and repositions the domains. When visualized in the molecular structure, the positioning of the domains of Aro1 matches the order of reactions in the pathway in a counter-clockwise fashion, with the domain that catalyses the first reaction ($Aro1_{DHQS}$) positioned at the "bottom" of the dimer, with the next two domains ($Aro1_{DHQase}$ and $Aro1_{DHSD}$) positioned sequentially above $Aro1_{DHQS}$, followed by $Aro1_{SK}$ at the "top left" and terminating with $Aro1_{EPSPS}$ at the "bottom left" (Fig 3C). However, each individual active site is more than 40 Å from each other and each face outwards from the core of the dimer (Fig 3C). We also do not observe any obvious clefts or tunnels between active sites.

## Impairment of specific enzymatic activities of Aro1 has a dramatic effect on *C. albicans* growth and viability

To test the essentiality of Aro1's individual enzymatic activities, we generated conditional expression strains in *C. albicans* in which mutations were introduced in a constitutively expressed allele of *ARO1*, whereas the other wild-type allele was transcriptionally repressible. First, we replaced the promoter of one allele of *ARO1* with a tetracycline-repressible promoter (*tetO*) such that addition of the tetracycline analog doxycycline (DOX) represses transcription of this allele of *ARO1*. In this strain, in the absence of DOX, *ARO1* was overexpressed at the transcript (Fig S8) and protein level (Fig S9), relative to the parental strain *C. albicans* SN95. To repress transcription of the ARO1 allele controlled by the tetO promoter, cultures were incubated overnight in a low concentration of DOX, then subcultured into media with (or without) a high concentration of DOX. Based on transcriptional analysis, the DOX treatment reduced *ARO1* transcript abundance to ~50% of wild-type levels (Fig S8), but did not alter growth on solid or liquid synthetic complete medium (Fig 4A and B). As anticipated, introduction of the *tetO* promoter at both alleles of *ARO1* resulted in further overexpression in the absence of DOX, whereas DOX reduced *ARO1* transcript abundance to ~1% of that observed for the wild-type strain (Fig S8) and caused a severe growth defect on solid and liquid medium (Fig 4A and B). After 24 h of transcriptional repression, the cells were examined by fluorescence microscopy. As anticipated, DOX treatment resulted in abnormal cell morphology and separation defects only for the *C. albicans tetO-ARO/tetO-ARO1* strain (Fig 4C). Furthermore, we observed a significant increase in cell death upon repression of ARO1, based on staining with the membrane-impermeable dye propidium iodide (Fig 4C).

To determine the essentiality of individual amino acids in the active sites of the five domains of Aro1 revealed by our structural analysis, we altered the remaining wild-type allele of *ARO1* in *C. albicans tetO-ARO1/ARO1*. To monitor expression of the mutant Aro1 derivatives, we introduced a TEV protease cleavable hexahistidine tag ($His_6TEV$) to the N-terminus of Aro1. To first confirm that this tag did not interfere with Aro1 activity, we introduced a *tetO* promoter and $His_6TEV$ tag at both *ARO1* alleles in *C. albicans*. Similarly to *C. albicans tetO-ARO1/tetO-ARO1*, *C. albicans tetO-His_6TEVARO1/tetO-His_6TEVARO1* had a growth defect on solid (Fig 4A) and in liquid medium (Fig S12) when doxycycline was added. Overexpression and DOX-dependent depletion of $His_6TEVAro1$ was confirmed by Western blotting (Fig S9). This indicated that both alleles of *ARO1* were tagged and functional in vivo. We then generated a DNA construct in which $His_6TEVAro1$ was under the control of the constitutive *ACT1* promoter ($P_{ACT1}$). This construct was introduced into the wild-type allele of *C. albicans tetO-ARO1/ARO1*, creating *C. albicans tetO-ARO1/$P_{ACT1}$-His_6TEVARO1*. As expected, this strain did not present a growth defect on solid (Fig 4) or liquid medium (Fig S12), because of expression of functional $His_6TEV$-tagged Aro1 from the *ACT1* promoter. Next, we generated isogenic strains encoding site-specific amino acid substitutions, truncations, and internal domain deletions to $P_{ACT1}$-$His_6TEVARO1$. For deletion of the DHQS domain ($Aro1^{DHQSΔ}$) the $His_6TEV$ tag was fused to the N-terminus of $Aro1^{388-1551}$; for deletion of the DHSD domain ($Aro1^{DHSDΔ}$), a stop codon was introduced at $Ile^{879}$; and the three internal domains were deleted and replaced with a $(GlySerSer)_6$ linker ($Aro1^{EPSPSΔ::(GSS)6}$, residues 387–857; $Aro1^{SKΔ::(GSS)6}$, residues 849–1,040; $Aro1^{DHQaseΔ::(GSS)6}$, residues 1,036–1,284). All truncation and deletion containing Aro1 derivatives appeared at expected molecular weight and at equal abundance when compared with the wild-type $His_6TEVAro1$ based on Western blotting of *C. albicans* whole-cell lysates (Fig S9); $Aro1^{DHQSΔ}$ appeared at greater abundance. Introduction of amino acid substitutions did not alter protein abundance in Western blots (Fig S9).

DOX-dependent depletion of *tetO-ARO1* in strains carrying $His_6TEVAro1$ with predicted inactivating substitutions in the DHQS domain of Aro1 as informed by structural analysis ($Aro1^{H264K}$, $Aro1^{H280K}$, and $Aro1^{DHQSΔ}$) conferred severe DOX-dependent growth defects on solid (Fig 4A) and liquid medium (Fig S12). Almost all cells had abnormal morphology and were dead after 24 h growth in liquid medium with DOX, based on fluorescence micrographs (Fig S10). The same growth phenotype was observed for EPSPS domain mutants ($Aro1^{D715A}$, $Aro1^{H796A}$, and $Aro1^{EPSPSΔ::(GSS)6}$) (Figs 4A and S12), except no cell death was evident in DOX-treated cultures after 24 h (Fig S10). SK mutants ($Aro1^{D890A}$, $Aro1^{R980A}$, and $Aro1^{SKΔ::(GSS)6}$) had a severe growth defect on solid medium (Fig 4A), and reduced growth in liquid medium (Fig S12); cells were rounded with a minor separation defect, and a minority of dead cells were identified. DHSD mutants ($Aro1^{K1334E}$, $Aro1^{D1370A}$, and $Aro1^{DHSDΔ}$) had reduced growth on solid (Fig 4A) and liquid medium (Figs 4A and S12); cells were enlarged, and a minority of dead cells were identified (Fig S10). No effects on growth or viability were identified for DHQase mutants ($Aro1^{E1072R}$, $Aro1^{R1194E}$, and $Aro1^{DHQaseΔ::(GSS)6}$).

We also investigated the effect of Aro1 interdomain interactions captured by our structural analysis. However, the amino acid substitutions at residues that mediate interdomain interactions

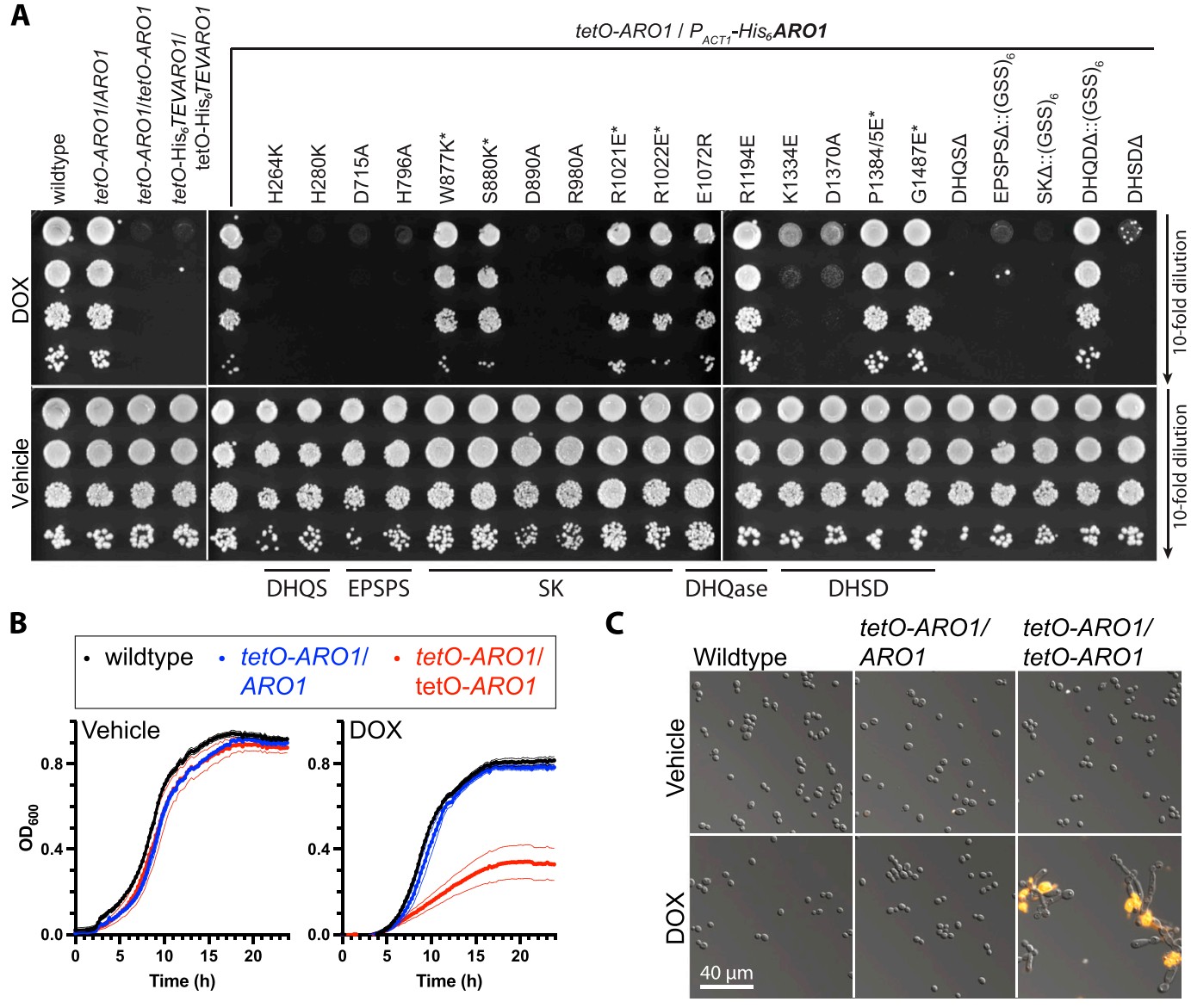

**Figure 4. Effect of Aro1 active site mutations on *C. albicans* viability.**
**(A)** *ARO1 activities are required for growth of C. albicans*. SC medium lacking arginine and supplemented with 150 µl NAT was inoculated with single colonies of *C. albicans* and incubated overnight at 30°C. Subcultures were prepared at 1:1,000 in SC medium supplemented with 50 ng/ml DOX and incubated overnight at 30°C. Cultures were normalized to 10 OD600 units/ml then 10-fold serial diluted in water. Diluted cultures were spotted onto SC Agar and SC Agar supplemented with 5 µg/ml DOX, then photographed after 48 h incubation at 30°C. Asterisk indicates substitutions to amino acids identified to mediate interdomain contacts. **(B)** *ARO1 is required for growth of C. albicans*. SC medium lacking arginine and supplemented with 150 µl NAT was inoculated with single colonies of *C. albicans* and incubated overnight at 30°C. 80 µl SC cultures were prepared at 0.001 OD600 unit/ml, in the presence and absence of 5 µg/ml DOX, then incubated at 30°C for 24 h. Data are mean optical densities for quadruplicate cultures ± SD for a representative experiment. **(B, C)** *ARO1 is required for viability of C. albicans*. Representative fluorescence micrographs of cultures from (B), stained with 1 µg/ml propidium iodide to label inviable cells.

(Aro1$^{W877K}$, Aro1$^{S880K}$, Aro1$^{R1021E}$, Aro1$^{R1022E}$, Aro1$^{P1384EP1385E}$, and Aro1$^{G1487E}$) showed no effect on growth or viability of corresponding *C. albicans* strains (Fig 4A).

## Enzymatic characterization of Aro1 active site mutants

To define how site-specific substitutions in each of the five active sites of Aro1 affect their enzymatic activities, we tested the five enzymatic domains of this multi-domain protein in vitro using previously established protocols (see the Materials and Methods section for details). In contrast to wild-type Aro1, the version of this protein carrying H268K/H280K double mutation in the N-terminal DHQS domain was successfully expressed and purified from *E. coli* as an N-terminal His$_6$TEV-tag fusion. Aro1$^{H268K/H280K}$ was catalytically inactive in a DHQS activity assay as compared with wild-type His$_6$TEV-tagged Aro1 purified from *C. albicans* (Table 1). This result was consistent with previous studies that showed mutation to His residues involved in divalent metal ion coordination resulted in a

**Table 1. Relative enzymatic activity of Aro1 active site mutants.**

| Enzymatic activity relative to WT Aro1 | Active Site Mutant | | | | | | |
|---|---|---|---|---|---|---|---|
| | DHQS | EPSPS | | SK | | SDH | |
| | H268K/H280K | D715A | H796A | D890A | R980A | K1334E | D1370A |
| | N.d. | 1.6% | 20.6% | 1.9% | 2.9% | N.d. | N.d. |

n.d. none detected.

complete loss of enzymatic activity in an AroM protein (Park et al, 2004). This suggested that the severe growth defect we observed for the *C. albicans* strain expressing this Aro1 mutant is due to a complete loss of one of the individual enzymatic activities of this protein (Fig 4A).

Next, we used the expression construct for Aro1$^{H268K/H280K}$ as a base for introduction of mutations in other Aro1 domain active sites. Catalytic activity in other Aro1 domains was unaffected by this mutation (Fig S11). As above, the corresponding Aro1 mutants were purified from *E. coli* and their enzymatic activities were compared to wild-type His$_6$TEV-tagged Aro1 purified from *C. albicans* (Table 1). We found that active site mutations in SK and DHSD domains resulted in near-complete and complete losses of corresponding activities for Aro1, respectively. Similarly, a near-complete loss of activity was observed for the D715A mutant localised to the EPSPS domain. Introduction of the H796A mutation in this domain had a less severe effect, with Aro1 retaining ~20% of EPSPS activity relative to the wild-type protein. Overall, our enzymatic analysis demonstrated that selected active site mutations severely abrogated the DHQS, EPSPS, SK, and DHSD activities of *C. albicans* Aro1 multi-domain enzyme. Taken together with the growth defect phenotypes that we observed for *C. albicans* strains expressing the corresponding Aro1 active site mutants, these results suggested that the enzymatic activity of each of these four Aro1 domains is critical for *C. albicans* growth and viability.

### The DHQase domain of Aro1 is catalytically inactive

Our structural analysis revealed significant deviation in active site composition of the Aro1 DHQase domain and the *C. albicans* strain expressing an Aro1 derivative lacking this domain (Aro1$^{DHQaseΔ::(GSS)6}$) showed no detectable growth defect in contrast to other *ARO1* mutants (Figs 4A and S12). These data led us to hypothesize that the DHQase domain of Aro1 in *C. albicans* is catalytically inactive. To investigate this, we purified wild-type Aro1 from *C. albicans tetO-His$_6$TEVARO1/tetO-His$_6$TEVARO1* and tested it for DHQase activity in vitro (Fig 5A). The Aro1 enzyme showed no detectable activity in this assay, whereas the *E. coli* type I DHQase, AroD, which we used as a control had 1.3 × 10$^3$ U/mg DHQase activity under the same conditions. Because Aro1 is a multi-domain enzyme, we considered the possibility that the 3-dehydroshikimate product measured in our assay may have been rapidly converted to shikimate via the activity of the DHSD domain. In such a scenario, our assay may have failed to capture product formation. To account for this, we expressed and purified the isolated DHQase domain of Aro1 (residues 1,041–1,249). Consistent with full-length Aro1, we observed no DHQase activity for this fragment (Fig 5A).

A catalytic His is conserved in type I DHQases and its substitution to Ala in *E. coli* AroD severely abrogated DHQase activity (Leech et al, 1995). Because *C. albicans* Aro1 features a Gln at this position, we sought to determine if this substitution (His to Gln) would eliminate enzymatic activity in a functionally active DHQase. Indeed, the *E. coli* AroD$^{H143Q}$ completely lost DHQase activity in our assay (Fig 5) in line with our hypothesis that DHQase domain in *C. albicans* Aro1 carries active site substitutions incompatible with catalytic activity typical for this class of enzymes.

Next, we performed a comparative analysis of Aro1 sequences across seven clinically important *Candida* species including *C. albicans*, *C. auris*, *Candida glabrata*, *Candida lusitaniae*, *Candida parapsilosis*, *Candida tropicalis*, and *Candida. viswanathii* (Fig 5B). All seven Aro1 sequences showed conservation in all residues established to be essential for activity of the four *C. albicans* Aro1 enzymatic domains described above; however, five of seven Aro1 sequences analysed showed substitution to the DHQase domain's catalytic His residue, whereas three of the seven also contained substitutions at the position corresponding to catalytically essential Lys residue in this domain. Only the *C. glabrata* and *C. parapsilosis* Aro1 sequences retained the conserved catalytic dyad (Fig 5B). Based on this analysis we predicted that these two Aro1 enzymes feature catalytically active DHQase domains. To test this, we expressed and purified *C. glabrata* and *C. parapsilosis* Aro1 proteins as N-terminal His$_6$TEV fusions in *E. coli*. Both multi-enzymes showed robust DHQase activities of 23.7 U/mg and 22.3 U/mg, respectively. Thus, our analysis revealed an important divergence in activity of Aro1 among *Candida* species, with the multi-domain enzyme in *C. albicans* showing no DHQase activity in contrast to its homologues found in *C. glabrata* and *C. parapsilosis*.

The discovery that *C. albicans* Aro1 does not have DHQase activity prompted us to search the genome for genes encoding other enzymes that might perform this reaction. We did not identify any genes encoding type I DHQase homologs, so we searched for type II DHQases, which perform the same reaction but share no sequence or structural similarity and use a different catalytic mechanism (Roszak et al, 2002). We identified *DQD1*, which encodes a type II DHQase homolog that adopts the canonical fold and dodecamerization expected for type II DHQases (Trapani et al, 2010). Notably, *DQD1* was identified as an essential gene in *C. albicans* by the GRACE strategy (Fu et al, 2021) and by mapping transposon insertions in a stable haploid isolate of *C. albicans* (Segal et al, 2018). Purified Dqd1 showed 1.5 × 10$^3$ U/mg DHQase activity in vitro, which is comparable with that observed for *E. coli* AroD and higher than DHQase activity we observed for *C. glabrata* and *C. parapsilosis* Aro1 enzymes (Fig 5A). This suggests that the activity of Dqd1 in *C. albicans* may be

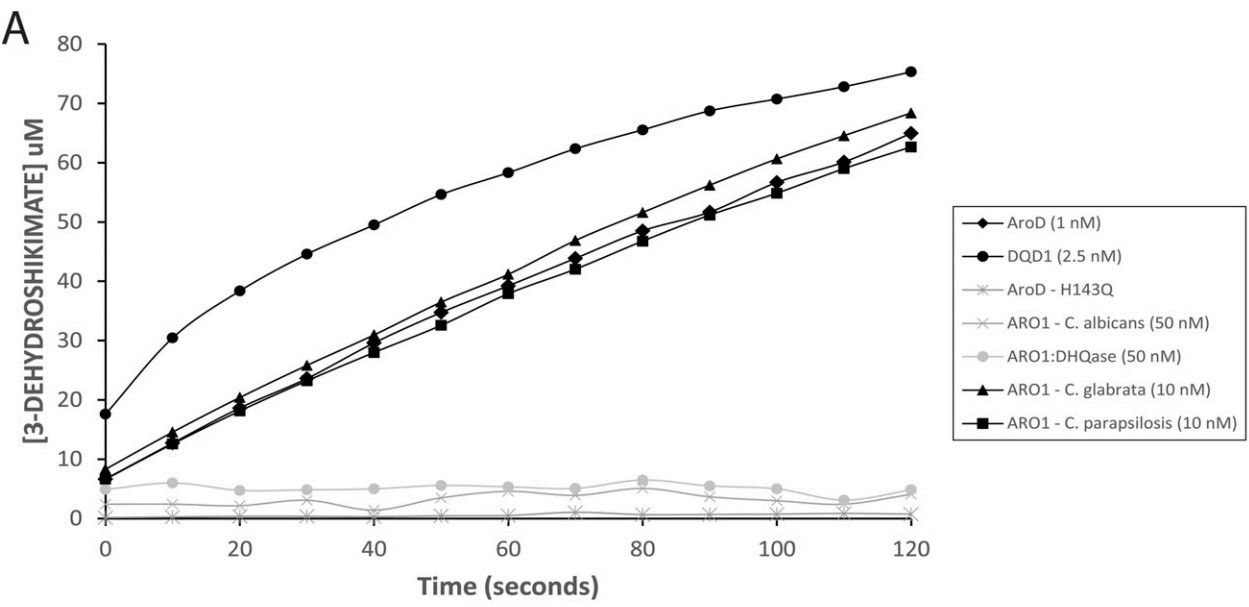

Figure 5. **Diversity of type I DHQase active site composition in *Candida* species.**
**(A)** *C. albicans* Aro1 is catalytically inactive in a DHQase activity assay. *Candida glabrata* and *C. parapsilosis* Aro1 were active, along with *C. glabrata* Dqd1. *E. coli* AroD was included as a positive control. **(B)** Multiple sequence alignment of the region of DHQase containing residues that comprise the catalytic dyad. Positions where residues involved in catalysis are expected are denoted with a (*), and the position where the catalytic His is expected is highlighted in blue. If a DQD1 ortholog was found in the genome, it is listed on the right and given as percent shared amino acid sequence identity.

sufficient to support the shikimate pathway and proposes a new role for this enzyme. We then searched each of the clinically relevant *Candida* genomes mentioned above for the presence of a gene encoding a type II DHQase. In all instances where mutations have occurred in the catalytic dyad of the Aro1$_{DHQase}$ domain, there was a DQD1 ortholog present in the genome; however, no type II DHQase was found in the genome of either *C. glabrata* or *C. parapsilosis* (Fig 5B). This analysis demonstrates a correlation between the presence of a type II DHQase and the loss of DHQase activity in the Aro1 multi-domain enzyme.

## Discussion

AroM/Aro1 multi-enzymes that catalyse five consecutive reactions of the shikimate pathway are a distinctive feature of chorismate biosynthesis in apicomplexan and fungal species, such as *Candida*. Aro1 is essential for *C. albicans* viability and pathogenesis, opening the possibility to target its enzymatic activities in new antifungal therapies. This study uncovers the molecular features of *C. albicans* Aro1 that paves the way for design of such inhibitors.

To gain molecular insight into this protein's activities, we visualized *C. albicans* Aro1's domains and the full-length protein using a combination of protein crystallography and single-molecule cryo-oEM SPR approaches. These analyses revealed that *C. albicans* Aro1 forms a dimer, the oligomeric state shared by AroM from *C. thermophilum*; the only other fully structurally characterised representative of this multienzyme family (Verasztó et al, 2020). Our structural analysis suggests the Aro1 dimer is primarily maintained through interactions between N-terminal DHQS domains and supported by additional interactions between DHQase domains of the protomers. However, an Aro1 construct containing a deletion of the DHQase domain was able to support wild-type growth of *C. albicans*, suggesting that this domain is not required for folding of the rest of Aro1 protein into an active conformation(s).

Wheeras the Aro1 dimer is reminiscent of that observed in *C. thermophilum* AroM, Aro1 monomers adopt distinct conformations

in the dimer which were not observed in the structure of AroM. We refer to these distinct conformations of Aro1 chains in the dimer as the "tight" and "loose" monomers, respectively (Fig 3). Notably, the "tight" monomer does not permit direct interaction between the Aro1$_{SK}$ and Aro1$_{EPSPS}$ domains, despite cross-linking mass spectrometry data for *C. thermophilum* AroM suggesting that the AroM$_{SK}$ domain and AroM$_{EPSPS}$ subdomain II can co-localise in solution (Veraszto et al, 2020). This discrepancy may be due to significant interdomain movement observed by cryoEM SPR in truncated Aro1 fragments and also reported for the AroM protein. Accordingly, observed asymmetry in Aro1 protomers may represent stable intermediates within the conformational cycle of Aro1 that is also implied for the AroM protein.

The impact of assembly of multiple enzymatic domains into a single Aro1/AroM protein for their catalytic activity remains elusive. In vitro characterization of *C. thermophilum* AroM concluded that concatenation of domains catalysing consecutive enzymatic reactions into a single protein does not increase the flux through the pathway (Veraszto et al, 2020). As in *C. thermophilum* AroM, our cryo-EM SPR structure of *C. albicans* Aro1 did not reveal that the individual active sites are spatially co-localized, nor is there evidence of any structural features connecting them. These observations suggest that there is no exchange of substrates/products between active sites and concatenation of the domains in *C. albicans* Aro1 should not increase the flux through the pathway; however, this remains to be experimentally deduced. Given our observation that the DHQase domain in this multi-enzyme is inactive, there are clearly differences between fungal Aro1/AroM that remain to be explored. The relative orientation between the Aro1$_{SK}$ and Aro1$_{EPSPS}$ domains changes dramatically between "tight" and "loose" conformations of Aro1 protomers (Fig 3), which may facilitate diffusion of substrates between these domains and support flux between consecutive steps supported by these Aro1 domains. The "blurring" of the subdomain I of Aro1$_{EPSPS}$ in our cryoEM SPR analysis suggests local movement of this fragment. The exchange from "loose" to "tight" conformation may restrict the movement of this portion of Aro1$_{EPSPS}$ and promote the catalytically active positioning between the two Aro1$_{EPSPS}$ subdomains.

DHQS, EPSPS, SK, and DHSD domains of Aro1 (Fig 1) shared strong structural similarity with single-domain bacterial and fungal shikimate pathway enzymes, including in the molecular architecture of their active sites. This allowed us to identify key residues important for individual catalytic activities of Aro1. We constructed a collection of Aro1 mutants with impaired activity in one of the domains, which was verified by in vitro enzymatic assays. *C. albicans* strains expressing these Aro1 mutants demonstrated severe growth defects, particularly in those carrying active site substitutions in the DHQS and EPSPS domains. Notably, we observed a severe growth defect in the H796A mutant in the EPSPS domain despite this mutant retaining ~20% of corresponding activity in vitro, suggesting that this level of EPSPS activity is insufficient to support viability of *C. albicans* (Table 1). Strong growth defects and cell death were also observed in case of *C. albicans* strains carrying active site mutations in the SK and DHSD domains. The difference in severity of observed effects between the growth on solid and liquid media may be due to significant clumping of the cells due to cell lysis that would interfere with optical density reading used for the latter assay. These

data suggest that *C. albicans* lacks enzymes that could provide compensatory activities to support growth in the absence of the DHQS, EPSPS, SK, and DHSD activities of Aro1, at least under the cultivation conditions tested in this study. Combined, this data clearly demonstrates that the activity of each of these four Aro1 domains is essential for the viability of *C. albicans*. Whereas the effect of essentiality of these four individual domains of Aro1 remains to be tested in infection models, our data suggest an important novel direction for antifungal therapy can be directed to targeting of one of these four enzymatic activities of Aro1.

Unlike the other Aro1 domains, the DHQase domain in *C. albicans* Aro1 lacked the catalytic residues typical of homologous type I DHQase enzymes. We demonstrated that this domain was not active in vitro, both as part of Aro1 or when expressed as a single-domain fragment. We identified similar active site aberrations in several other *Candida* species, including *C. auris*, *C. lusitaniae*, *C. tropicalis*, and *C. viswanathii*. *C. glabrata* and *C. parapsilosis* Aro1 retained the typical DHQase active site composition and purified Aro1 enzymes from these organisms had robust DHQase activity in vitro. Because the deletion of the DHQase domain of *ARO1* did not have any effect on *C. albicans* viability, this raised the question of which enzyme was responsible for supporting this reaction in the shikimate pathway in these species. Our analysis identified that this function in *C. albicans* is fulfilled by the type II DHQase encoded by *DQD1*. We demonstrated that this essential gene product shows robust activity for conversion of 3-dehydroquinate to 3-dehydroshikimate, and this activity is even higher than that of corresponding domains in *C. glabrata* and *C. parapsilosis* Aro1 enzymes under the same assay conditions. Interestingly, these two *Candida* species do not encode for *DQD1* orthologs, whereas we identified *DQD1* orthologs in all *Candida* species encoding DHQase-inactive variants of Aro1. Type II DHQases are typically involved in quinate catabolism in fungi; however, the type II DHQase from the Gram-negative bacterium *Acidomonas methanolica* participates in both quinate catabolism and shikimate biosynthesis (Euverink et al, 1992). We hypothesize that the presence of a more efficient type II DHQase in *C. albicans* may have alleviated selective pressure for maintaining DHQase activity within Aro1. Similarly, the lack or loss of a type II DHQase ortholog in *C. glabrata* and *C. parapsilosis* necessitated maintenance of the catalytic dyad and consequently, enzymatic activity, in the DHQase domain of Aro1 in these species.

Because type I and type II DHQases are structurally and mechanistically distinct, the observation that *C. albicans*, and other clinically important *Candida* species such as *C. auris*, rely on a type II DHQase rather than the type I DHQase domain present in Aro1 informs on potential therapeutic strategies targeting this step of the shikimate pathway. Further analysis is warranted into other possible deviations in the shikimate pathway of other species.

# Materials and Methods

### Recombinant protein purification

*E. coli* BL21(DE3) Gold was used for overexpression of *C. albicans* Aro1 coding for residues 1–387 residues corresponding to the DHQS

domain (Aro1$_{DHQS}$); residues 387–844, corresponding to the EPSPS domain (Aro1$_{EPSPS}$); and the C-terminal residues 845–1,551, comprising the SK, DHQase, and DHSD domains (Aro1$_{SK-DHQase-DHSD}$). 1 liter LB media containing 100 $\mu$g/ml ampicillin was inoculated with 3 ml of overnight culture and incubated at 37°C with shaking. Cultures were induced with IPTG at 17°C once the OD$_{600}$ reached 0.6–0.8. Cell pellets were collected by centrifugation at 7,000$g$. Ni-NTA affinity chromatography was used for protein purification. Cells were resuspended in a binding buffer (100 mM Hepes, pH 7.5, 300 mM NaCl, 5 mM imidazole, and 5% glycerol [vol/vol]) and lysed with a sonicator. Then insoluble cell debris was removed by centrifugation at 30,000$g$. The soluble cell lysate fraction was loaded on a 4 ml Ni-NTA column (QIAGEN) pre-equilibrated with binding buffer, washed with 150 ml washing buffer (100 mM Hepes, pH 7.5, 300 mM NaCl, 30 mM imidazole, and 5% glycerol [vol/vol]), and N-terminal His6-tagged protein was eluted with elution buffer (100 mM Hepes, pH 7.5, 300 mM NaCl, 250 mM imidazole, and 5% glycerol [vol/vol]). The His$_6$-tagged proteins were then subjected to overnight TEV cleavage using 50 $\mu$g of TEV per mg of His6-tagged protein and dialyzed overnight against the binding buffer. The His-tag and TEV were removed by rerunning the protein over the Ni-NTA column. The tag-free protein was then dialyzed against crystallisation buffer (50 mM Hepes, pH 7.5, 300 mM NaCl) and the purity of the protein was analysed by SDS-polyacrylamide gel electrophoresis.

## Crystallization and X-ray structure determination

All crystals were grown at room temperature using the vapor diffusion sitting drop method. For the Aro1$_{DHQS}$-NADH crystal, 25 mg/ml protein was mixed with 2 mM NADH and 0.5 mM ZnCl$_2$, and then with reservoir solution 0.2 M sodium thiocyanate and 20% (wt/vol) PEG3350; for the Aro1$_{EPSPS}$-shikimate-3-phosphate crystal, 25 mg/ml protein was mixed with 1 mM shikimate-3-phosphate, and then with reservoir solution 0.4 M sodium chloride, 0.1 M Tris, pH 8, and 26% (wt/vol) PEG3350; for the Aro1$_{SK-DHQase-DHSD}$ crystal, 25 mg/ml protein was mixed with 2 mM shikimate-3-phosphate, 1 mM ATP, 0.1 M sodium formate, and 12% (2/v) PEG3350. Cryoprotection solutions for the Aro1$_{DHQS}$, Aro1$_{EPSPS}$ and the Aro1$_{SK-DHQase-DHSD}$ crystals were: 8% glycerol, 8% ethylene glycol, 8% sucrose; 22% ethylene glycol, Hepes, pH 7.8; and 20% glycerol plus 4% tacsimate.

Diffraction data at 100 K were collected at beamline 19-ID of the Structural Biology Center at the Advanced Photon Source, Argonne National Laboratory. Diffraction data were processed using HKL3000 (Minor et al, 2006). Structures were solved by Molecular Replacement using Phenix.phaser and the following models: DHQS domain from AroM from *A. nidulans* (PDB 1SG6), (Nichols et al, 2004a); EPSPS from *V. cholerae* (PDB 3NVS); DHQase+DHSD domains from *S. pombe* AroM (PDB 5SWV). Both model building and refinement were performed using Phenix.refine (Afonine et al, 2012) and Coot (Emsley et al, 2010). Atomic coordinates have been deposited in the Protein Data Bank with accession codes 6C5C, 7TBU and 7TBV. Structural homologs were identified in the PDB using the Dali-lite server (Holm, 2020).

## Dehydroquinate synthase (DHQS) assay

DHQS reactions were performed in 250-$\mu$l volumes containing 50 mM Tris–HCl, pH 7.5, 20 nM NAD+, 0.2–0.8 mM DAHP, and 10 nM of Aro1 or Aro1 mutant protein. Enzyme activity was allowed to proceed for 5 min at room temperature (22°C), after which the reaction was terminated via the addition of 100 $\mu$l of malachite green assay solution (Echelon Biosciences). After incubation for 30 min post termination, absorbance was read at 620 nm using a Victor 1,420 plate reader (Perkin-Elmer), and the levels of inorganic phosphate were determined by comparison to a phosphate standard curve generated according to the manufacturer's protocol.

## EPSP synthase assay

EPSPS activity was assayed at 22°C in a reaction buffer containing 50 mM Hepes, pH 7.5, 100 mM KCl, 2 mM DTT, 1 mM shikimate-3-phosphate, and 1 mM phospoenol pyruvate. Aro1 and Aro1 mutant protein was added to a final concentration of 10 nM, and the reaction was allowed to proceed for 5 min, then terminated with the addition of 200 $\mu$l of malachite green assay solution (Echelon Biosciences). The remainder of the procedure was performed as described above for the DHQS assay. Enzyme activity was expressed in terms of production of P$_i$ over time, and the activity of Aro1 mutants was expressed relative to wild-type Aro1.

## Shikimate kinase assay

SK activity was measured using the ADP-Glo Kinase Assay kit (Promega). The assay was performed in a 25-$\mu$l volume containing 100 mM Tris–HCl, pH 7.5, 10 $\mu$M ATP, 20 mM MgCl$_2$, and 100 $\mu$M shikimic acid. Catalysis was initiated by the addition of Aro1 or Aro1 mutant protein to a final concentration of 10 nM and allowed to proceed for 30 min at 22°C, then terminated via the addition of 25 $\mu$l of ADP-Glo Reagent. After 40 min of incubation, 50 $\mu$l of the kinase detection reagent was added, samples were incubated for 30 min, and luminescence was quantified using a Victor 1,420 plate reader (Perkin-Elmer). The amount of ATP converted to ADP by Aro1 was determined by comparing luminescence readings to a standard curve generated according to the manufacturer's instructions.

## Shikimate dehydrogenase assay

The assay for shikimate dehydrogenase activity was performed as described by Kubota et al (2013) and measured the conversion of NADPH to NADP+ by reading absorbance at 340 nm ($\varepsilon$ = 6.2 × 10$^4$ M$^{-1}$ cm$^{-1}$). All reactions were performed using a final volume of 1 ml containing 100 mM Tris–HCl, pH 7.5, 200 $\mu$M NADPH, and 500 $\mu$M 3-dehydroshikimate (MilliporeSigma). The reaction was initiated by adding the Aro1 enzyme to a final concentration of 5 nM, then monitored for 2 min at 37°C.

## DHQase assay

DHQase activity was tested as previously described by Chaudhuri et al (1986). Briefly, formation of 3-dehydroshikimate was measured by reading absorbance at 234 nm ($\varepsilon$ = 1.2 × 10$^4$ M$^{-1}$ cm$^{-1}$). The assay was performed in 1-ml quartz cuvettes at 37°C in a buffer containing 100 mM Tris–HCl, pH 7.5, and 0.5 mM 3-dehydroquinate. The assay was initiated by the addition of protein and monitored for 2 min. Protein concentration ranges tested were *C. albicans* Aro1 (1–50 nM), *E. coli*

AroD (1–25 nM), *C. albicans* Dqd1 (0.25–10 nM), *C. glabrata*, and *C. parapsilosis* Aro1 (1–10 nM).

## Purification of Aro1 from *C. albicans*

*C. albicans* CaLC6830 was grown overnight at 30°C, 200 rpm to late log phase, then harvested via centrifugation (15 min, 5,000*g*, 4°C). The cell pellet was flash frozen in liquid $N_2$, and then ground under liquid $N_2$ using a mortar and pestle until the consistency reached a fine powder. The powder was resuspended in Binding Buffer (300 mM NaCl, 50 mM Hepes, pH 7.5, 5 mM imidazole, and 5% glycerol). $His_6$-tagged Aro1 was then purified using Ni-NTA chromatography. After purification, the sample was incubated overnight at 4°C with TEV protease to remove the 6His tag and passaged over a second Ni-NTA column. The flowthrough was concentrated and fractionated using a Superdex 200 column (150 mM NaCl and 10 mM Hepes, pH 7.5). The eluate was concentrated and flash frozen in liquid $N_2$ for cryo-EM analysis, or diluted with glycerol (final concentration 50%) and stored at −20°C for use in enzymatic assays.

## Cryo-EM grid preparation, data collection, and image processing

Quantifoil R1.2/1.3 300 mesh Au grids were glow-discharged with air at 30 mA for 80 s in a PELCO easiGlow Glow Discharge Cleaning System. Freshly glow-discharged grids were put into an FEI Vitrobot Mark IV at 4°C and 100% humidity. Aro1 protein was thawed and 3 $\mu$l of sample at 2.2 or 0.55 mg/ml was applied to the grids. Grids were blotted for 5 s before vitrification by plunging into liquid ethane cooled by liquid $N_2$. Vitrified samples were stored in liquid $N_2$ until imaging. Movies were acquired on a 300 keV Titan Krios G2 microscope (Thermo Fisher Scientific) using a K3 Summit direct electron detector (Gatan) in super-resolution mode with a nominal magnification of 1,050,00× and pixel size of 0.4165 Å/pixel. SerialEM was used for automated data collection in beam-image shift mode with nine images collected per stage movement and a defocus range from −1.5 to −3 $\mu$m. The slit width of the GIF Quantum Energy Filter was set to 25 eV. Movies were dose-fractionated into 100 or 120 frames with a total dose of 83.5 or 72.4 electrons/$\text{Å}^2$, respectively.

All movies were imported into CryoSPARC for motion correction using patch motion correction and binned to a pixel size of 0.833 Å/pixel followed by patch CTF estimation. Post-processing of all maps was performed using DeepEMhancer. Processing was performed on an initial dataset consisting of 932 manually curated micrographs. Initial particle picking was performed using Topaz autopicking with the pretrained model, resulting in 334,664 particles. Particles were extracted with a box size of 600 pixels and Fourier cropped to a pixel size of 2.50 Å. Multiple rounds of 2D classification were performed to identify 158,808 high quality particles. Ab-initio model building was performed in C1 symmetry using four classes to sort out broken and full-length particles, leading to one high quality class representing the dimer of the Aro1$_{DHQS}$ and Aro1$_{EPSPS}$ domain. A second dataset consisting of 1,623 high quality movies was collected and template-based autopicking was performed using templates generated from the Aro1$_{DHQS}$ and Aro1$_{EPSPS}$ dimer map. Particles were extracted in 600 pixel boxes and Fourier cropped to 400 pixels with a pixel size of 1.2495 Å/pixel. Multiple rounds of 2D classification were performed and the high

quality particles were combined with the particles from the previous dataset, resulting in 740,854 total particles. The particles were exported to star format using Csparc2star.py for further classification in Relion 3.1. This group of particles was subjected to 3D classification with eight classes using the Aro1$_{DHQ}$ and Aro1$_{EPSPS}$ dimer map as the initial model. After classification, one class consisting of 50,390 particles with features consistent with the full-length Aro1 dimer were identified. Another round of 3D classification with 10 classes was performed on the full 740,854 particle dataset using the Aro1 full-length dimer as the initial model. After classification, one class of 63,494 particles was imported into cryoSPARC for homogeneous refinement, resulting in a map of the full-length Aro1 protein at 4.52 Å resolution.

The particle set resulting in the 4.52 Å map was relatively inhomogeneous after 2D classification, so the dataset was reclassified using heterogeneous refinement in cryoSPARC. Heterogeneous refinement was run on the 740,854 particle dataset using the Aro1$_{DHQS}$ and Aro1$_{EPSPS}$ dimer map, the full-length Aro1 map, and two noise maps generated from ab-initio reconstruction until the noise classes contained less than 5% of the total particles. A total of 87,484 particles corresponded to the full-length dimer of Aro1 and after homogeneous refinement the final map resolution was 4.16 Å. Tight masks were generated around the Aro1$_{SK-DHQase-DHSD}$ dimer and Aro1$_{DHQS}$ and Aro1$_{EPSPS}$ dimer for local refinement in cryoSPARC. Local refinement of the Aro1$_{SK-DHQase-DHSD}$ dimer and Aro1$_{DHQS}$ and Aro1$_{EPSPS}$ dimer resulted in 3.16 and 3.43 Å maps, respectively. The particles from the final homogeneous refinement were exported to Relion 3.1 and subjected to focused classification without image alignment with masks around Aro1$_{EPSPS}$ domain 1 from subunit 1 and subunit 2. After classification particles were exported to cryoSPARC and local refinement was performed with a mask around Aro1$_{DHQS-EPSPS}$ of subunit 1 and subunit 2, leading to maps at final resolutions of 4.23 and 4.63 Å, respectively.

## Cryo-EM model building full-length Aro1 dimer

The crystal structures for the Aro1$_{DHQS}$ (PDB ID: 6C5C) and Aro1$_{EPSPS}$ domains (PDB ID: 7TBU) were rigid body fit into the maps Aro1$_{DHQS-EPSPS}$ from subunit 1 and subunit 2. The monomer models were then rigid body fit into each protomer of the symmetric map of the Aro1$_{DHQ}$ and Aro1$_{EPSPS}$ dimer from the broken particles using Chimera. The dimer map was then fit into the focused map of the Aro1$_{DHQS}$ and Aro1$_{EPSPS}$ domains from the full-length particles using Chimera. The crystal structure of Aro1$_{SK-DHQ-DHSD}$ was a rigid body fit into the focused map of the Aro1$_{SK-DHQ-DHSD}$ dimer from the full-length particles using Chimera. Model building was performed on each focused map using Coot with real space refinement performed with Phenix.

## Alignment of *ARO1* orthologs

Aro1 sequences from *C. auris* (acc. No. XP_018172187.1), *C. glabrata* (acc. No. XP_449840), *C. lusitaniae* (acc. No. OVF04495.1), *C. parapsilosis* (acc. No. CCE44200), *C. tropicalis* (acc. No. XP_002545280.1), *C. viswanathii* (acc. No. RCK59526.1) were obtained from GenBank and aligned using Muscle (Edgar, 2004).

## Cloning of recombinant *AROD, DQD1, C. glabrata ARO1,* and *C. parapsilosis ARO1*

*AROD* (acc. no. NP_416208) was PCR-amplified from *E. coli* gDNA using Phusion polymerase (NEB). *C. albicans DQD1* (acc. no. XP_714872), *C. glabrata ARO1* (acc. no. XP_449840), and *C. parapsilosis ARO1* (acc. no. CCE44200) were codon-optimized for *E. coli* expression and synthesized (Codex DNA). All genes were cloned into the pMCSG53 expression vector via Gibson assembly (Codex DNA).

## Expression and purification of AroD, Dqd1, *C. glabrata* Aro1, and *C. parapsilosis* Aro1

Plasmids were transformed into the *E. coli* strain BL21(DE3)-Gold. Cells were grown at 37°C and 200 rpm to an OD600 of 0.8, cooled to 20°C then induced with 1 mM IPTG and incubated overnight. Cells were collected via centrifugation at 5,000$g$, resuspended in binding buffer (300 mM NaCl, 50 mM Hepes, pH 7.5, 5 mM imidazole, 5% glycerol) then sonicated. Proteins were purified using Ni-NTA chromatography, and treated with TEV protease after elution for removal of the His$_6$ tag.

## *C. albicans* strain construction

*C. albicans* strains used in this work are summarized in Table S3. Liquid cultures were grown in YPD (1% [wt/vol] yeast extract, 2% [wt/vol] bactopeptone, 2% [wt/vol] glucose), or synthetic defined (SD) medium (1.74 g/l yeast nitrogen base without amino acids without ammonium sulfate [BioShop], 2% [wt/vol] glucose, and 0.1% [wt/vol] sodium L-glutamate [Millipore Sigma]), or synthetic complete (SC) medium (SD medium supplemented with 1.57 g/l SC-Arg-His-Leu-Ura powder [Sunrise Science Products], 85.6 g/l L-arginine, 85.6 g/l L-histidine, 173.4 g/l L-leucine, and 85.6 g/l uridine [Millipore Sigma]). For solid medium, Agar was added to 2% (wt/vol). For selections with *SAT1*, nourseothricin (NAT; Jena Bioscience GmbH) was supplemented to 150 µg/ml. DNA constructs were PCR-amplified using phusion polymerase (NEB) using custom oligonucleotide primers (Thermo Fisher Scientific) described in Table S4. PCR products were cleaned using QIAquick PCR Purification Kit (QIAGEN) or purified using QIAquick Gel Extraction Kit (QIAGEN). Genomic DNA was purified using the PureLink Genomic DNA Mini Kit (Invitrogen). Overnight YPD cultures of *C. albicans* cultures were diluted 1:1 in 50% (wt/vol) glycerol and archived at −80°C. Recombinant plasmids used in cloning are described in Table S5 and were maintained in *E. coli* Top10.

### CaLC5759 and CaLC5761

An integrating repair cassette encoding *frt-P$_{SAP2}$-FLP-SAT-frt-tetO* with 72 bp 5′ homology to 250 bp upstream of the *ARO1* ORF, and 3′ homology to the first 72 bp of the ARO1 ORF was PCR-amplified from pLC605 using oligonucleotide primers oLC7059/oLC7060. A cassette for transient expression of an sgRNA targeting *P$_{ARO1}$* was constructed by fusion of two PCR products amplified from plasmid pLC963 (Veri et al, 2018). Product 1 was amplified using oLC5979/oLC7048, and Product 2 using oLC7047/oLC5981. These PCR products were gel-purified and then fused by PCR using primers oLC5979/

oLC5981. Integration of the *frt-P$_{SAP2}$-FLP-SAT-frt-tetO* construct eliminates 250 bp of *P$_{ARO1}$* including the sgRNA target sequence. A cassette for transient expression of Cas9 was amplified from plasmid pLC963 using oLC6924/oLC6925. The three PCR constructs were transformed into *C. albicans* SN95 (Noble & Johnson, 2005) and transformants were selected by 2 d growth at 30°C on YPD NAT agar. Downstream integration of the *tetO-ARO1* construct was confirmed by colony PCR using oLC274/oLC7074. Presence of the wild-type junction between *P$_{ARO1}$* and *ARO1* was tested by colony PCR with oLC7071/oLC7074. CaLC5759 was heterozygous for integration of the tetO-ARO1 construct and CaLC5761 was homozygous.

### CaLC6830

An integrating repair cassette encoding *frt-P$_{SAP2}$-FLP-SAT-frt-tetO-His$_6$TEV* with 72 bp 5′ homology to 250 bp upstream of the *ARO1* ORF, and 3′ homology to the first 72 bp of the *ARO1* ORF was PCR-amplified from pLC605 using oligonucleotide primers oLC7059/oLC8510 (Veri et al, 2018). Cas9 and sgRNA transient expression cassettes were prepared as described for CaLC5759. The three PCR products were transformed into CaLC239, and transformants were selected by 2 d growth at 30°C on YPD NAT agar. Absence of the wild-type junction between *P$_{ARO1}$* and *ARO1* was confirmed by colony PCR with oLC7071/oLC7074. Downstream integration of the *tetO-ARO1 construct* was confirmed by colony PCR using oLC274/oLC7074. Protein expression was confirmed by Western immunoblotting of whole-cell lysates.

### CaLC7800, CaLC7801, and CaLC7803

An integrating repair cassette encoding *frt-P$_{SAP2}$-FLP-SAT-frt-tetO* with 72 bp 5′ homology to 250 bp upstream of the *DQD1* ORF and 3′ homology to the first 72 bp of the *DQD1* ORF was amplified from pLC605 using oLC9711/oLC9712. The PCR product was transformed into *C. albicans* SN95 and transformants were selected by 2 d growth at 30°C on YPD NAT agar. Downstream integration of the *tetO-DQD1* construct was confirmed by colony PCR using oLC274/oLC9714. Presence of the wild-type junction between *P$_{DQD1}$* and *DQD1* was identified by colony PCR using oLC9713/oLC9714. This heterozygous strain was designated CaLC7800. A single colony of CaLC7800 was used to inoculate 5 ml of SD medium lacking glucose and supplemented with 0.4% (wt/vol) bovine serum albumin and 2% maltose, to induce expression of FLP recombinase and drive excision of *SAT1*. After 48 h incubation at 30°C, cells were diluted in water and ~100 cells plated on YPD Agar. After 2 d incubation at 30°C, NAT sensitive colonies were identified by replica plating onto YPD NAT agar. *SAT1* excision was confirmed by colony PCR using oLC275/oLC9714. The resultant strain was designated CaLC7801. CaLC was then transformed with the same *frt-P$_{SAP2}$-FLP-SAT-frt-tetO* cassette to introduce the tetO promoter at the second *DQD1* allele. Absence of the wild-type junction between *P$_{DQD1}$* and *DQD1* was confirmed by colony PCR using oLC9713/oLC9714.

### CaLC6598

An integrating repair cassette encoding *P$_{ACT1}$*-His$_6$TEV*ARO1-ARG4* with 70 bp 5′ homology to 250 bp upstream of the *ARO1* ORF and 3′

homology to the first 70 bp after the *ARO1* ORF was constructed from three PCR products. First, $P_{ACT1}$ with 5′ homology to $P_{ARO1}$ was PCR-amplified from pLC620 using oLC7384/oLC7383 (Shapiro et al, 2012). Second, $His_6TEVAro1$ was amplified from pLC1335 using oLC7385/oLC7482. Third, a PCR product containing the *ARG4* selectable marker with 3′ homology to the ARO1 locus was amplified from pLC576 using oLC7387/oLC738 (Lavoie et al, 2008). These three PCR products were gel-purified, then 0.35 pmol of each was combined and fused using NEBuilder HiFi DNA Assembly Kit according to the manufacturer's instructions (New England Biolabs). The fusion product was amplified using primers oLC7384/oLC7388, then gel-purified. Transient expression cassettes for Cas9 and the $P_{ARO1}$-targeting sgRNA were prepared as described for CaLC5759. The three PCR products were transformed into CaLC5759, and transformants were selected by 2 d growth at 30°C on SC agar lacking arginine and supplemented with 150 µg/ml NAT. Absence of the wild-type junction between $P_{ARO1}$ and *ARO1* was confirmed by colony PCR with oLC7071/oLC7074, upstream integration of the $P_{ACT1}$-$His_6TEVARO1$-*ARG4* cassette with oLC7073/oLC1118, downstream integration of the $P_{ACT1}$-$His_6TEVARO1$-*ARG4* cassette with oLC4879/oLC7072, presence of the junction between *ARO1* and *ARG4* with oLC7075/oLC5893, and presence of the *tetO-ARO1* construct with oLC274/oLC7074. The $His_6TEVARO1$ allele was PCR-amplified from genomic DNA (PureLink Genomic DNA Mini Kit; Invitrogen) using oLC5385/oLC5893, and verified by Sanger sequenced using oLC7601, oLC7602, oLC7603, oLC7604, oLC7605, oLC7606, oLC7607, oLC7608, oLC7074, and oLC7076.

### CaLC6599

CaLC6599 was constructed as per CaLC6598, except $His_6TEVAro1^{388-1551}$ was amplified from pLC1349 using oLC7385/oLC7482.

### *C. albicans* strains encoding amino acid substitutions to, truncations to, or domain deletions within $His_6TEVARO1$

Isogenic strains to CaLC6598 were constructed using the following methodology. Construction of CaLC7016 is provided as a representative example. An integrating repair cassette was constructed encoding $P_{ACT1}$-$His_6TEVARO1^{H264K}$-*ARG4* with 70 bp of 5′ homology 250 bp upstream of the *ARO1* ORF and 70 bp 3′ homology immediately after the *ARO1* ORF. This construct was generated by amplification of two PCR products from purified CaLC6598 genomic DNA using oligonucleotide primer pairs oLC7384/oLC7109 (H264K-SDM-Rv) and oLC7108(H264K-SDM-Fw)/oLC7388. These PCR products were gel-purified and fused by PCR using primers oLC8966/oLC8967. For other mutant derivatives oLC7109 and oLC7108 were replaced with primers corresponding to the desired amino acid substitution or domain deletion described in Table S3. Transient expression cassettes for Cas9 and the $P_{ARO1}$-targeting sgRNA were prepared as described for CaLC5759. The three PCR products were transformed into CaLC5759 and transformants were selected by 2 d growth at 30°C on SC agar lacking arginine and supplemented with 150 µg/ml NAT. Absence of the wild-type junction between $P_{ARO1}$ and *ARO1* was confirmed by colony PCR with oLC7071/oLC7074, upstream integration of the $P_{ACT1}$-$His_6TEVARO1^{H264K}$-*ARG4* cassette with oLC7073/oLC1118, downstream integration of the $P_{ACT1}$-$His_6TEVARO1^{H264K}$-*ARG4* cassette with oLC4879/oLC7072, and presence of the *tetO-ARO1* construct with

oLC274/oLC7074. Genomic DNA was purified, the $His_6TEVARO1^{H264K}$ allele was PCR-amplified using oLC5385/oLC5893 and verified by Sanger sequencing using oLC7601, oLC7602, oLC7603, oLC7604, oLC7605, oLC7606, oLC7607, oLC7608, oLC7074, and oLC7076.

### pLC1335

$His_6TEV$-tagged *ARO1* was PCR-amplified from *C. albicans* SC5314 (Jones et al, 2004) genomic DNA using oLC7479/oLC7332. Both this PCR product and pBAD24 (Guzman et al, 1995) were digested with Sal1 and Sma1 (NEB), PCR purified, and then ligated using T4 DNA ligase (NEB). Plasmids RE mapped using EcoR1 and EcoRV (NEB) and then verified by Sanger sequencing using oLC7075, oLC7076, oLC7077, and oLC7385. QuikChange (Stratagene) site-directed mutagenesis with oLC8080/oLC8081 was used to correct a frameshift mutation in *ARO1* and silently removed a ClaI restriction site. This plasmid was verified by Sanger sequencing using oLC7601, oLC7602, oLC7603, oLC7604, oLC7605, oLC7606, oLC7607, oLC7608, oLC7074, and oLC7076.

### pLC1349

$His_6TEV$-tagged *C. albicans* $ARO1^{DHQSΔ}$ (amino acid residues 388–1,551) was PCR-amplified from pLC1335 using oLC7397/oLC7332. Both this PCR product and pBAD24 were digested with Sal1 and Sma1 (NEB), PCR purified, ligated using T4 DNA ligase (NEB), and transformed into *E. coli* Top10. Plasmid was verified by Sanger sequencing using oLC7602.

### qRT-PCR

10 ml liquid cultures were grown in SC medium overnight at 30°C, subcultured into 10 ml SC medium or SC medium supplemented with 0.05 µg/ml DOX overnight, then subcultured into 10 ml SC medium or SC medium supplemented with 5 µg/ml DOX for 4 h. Cells were collected by centrifugation at 3,000*g* for 10 min at 4°C, washed once with 1 ml of ice-cold water, then flash frozen in liquid $N_2$. RNA was extracted using RNeasy Mini Kit (QIAGEN). For cell lysis, the frozen pellet was resuspended in 650 µl ice-cold Buffer RLT supplemented with 10 µl/ml 2-mercaptoethanol, then disrupted with 0.5 mm acid-washed glass beads (Sigma-Aldrich) for 4 × 1 min using a Mini-BeadBeater-24 (BioSpec Products Inc.). Contaminating DNA was removed from 4 µg of the purified RNA using the DNA-free DNA Removal Kit (Invitrogen), according to the manufacturer's instructions. 1 µg of the DNAse-treated RNA was used to synthesize cDNA using the iScript cDNA Synthesis Kit (Bio-Rad). Transcript abundance was quantified from cDNA by real-time PCR using a CFX384 Touch Real-Time PCR Detection System (Bio-Rad), Fast SYBR Green Master Mix (Applied Biosystems), and oligonucleotide primers described in Table S4. Data were analysed using CFX Manager (Bio-Rad; v3.1). Amplicons were ~150 bp, had single peaks in melt curves, and abundances were within the linear range of six point 10-fold dilution standard curves.

### Growth assays

10 ml liquid cultures were grown in SC medium lacking arginine and supplemented with 150 µg/ml NAT with 200 rpm shaking overnight at 30°C then subcultured at 1:1,000 into 10 ml SC medium or SC

medium supplemented with 0.05 $\mu$g/ml DOX overnight. Cultures were diluted to an $OD_{600}$ of 1, then diluted at 1:80 into 96-well plates containing 80 $\mu$l/well SC medium or SC medium supplemented with 5 $\mu$g/ml DOX. Plates were sealed with an adhesive plate seal (AB-0580; Thermo Fisher Scientific), and placed in a custom-built growth curve robot (S&P Robotics Inc.). Within, cultures were incubated at 30°C and $OD_{600}$ was measured at 15 min intervals for 24 h.

## Microscopy

After growth curves described above, propidium iodide (Sigma-Aldrich) was added to cultures to 1 $\mu$g/ml (final). 5 $\mu$l of cell suspension was placed on a glass slide under a cover slip. Fluorescence micrographs were acquired on a Zeiss Axio Imager.M1 at 400 × magnification (40 × oil objective). Propidium iodide fluorescence was observed using an X-Cite series 120Q light source (Excelitas Technologies Corp.) and ET HQ DSRed (TRITC/Cy3) filter (Chroma Technology Corp.).

## Protein extraction and Western blotting

At mid-log phase, 10 ml cultures of *C. albicans* were collected by centrifugation at 3,000$g$ for 10 min at 4°C. Cell pellets were washed with 1 ml ice-cold water then flash frozen in liquid $N_2$. Whole-cell lysates were prepared by mild alkaline lysis, trichloroacetic acid precipitation, and solubilized in SDS/urea loading buffer as previously described, other than omission of bromophenol blue dye. Protein concentration was roughly normalized by absorbance at 280 nm. Lysates were separated in NuPAGE 3–8% acrylamide Tris-acetate gels using NuPAGE Tris-acetate SDS running buffer (Thermo Fisher Scientific). Proteins were transferred to PVDF membranes (Bio-Rad Immun-Blot, 0.2 $\mu$m). Membranes were blocked with 3% (wt/vol) bovine serum albumin in Tris-buffered saline with 0.5% (vol/vol) Tween-20 (TBS-T). Membranes were probed with mouse–anti-His$_5$ monoclonal primary antibody (QIAGEN; 1:3,000) and horseradish peroxidase-conjugated goat-anti mouse secondary antibody (Bio-Rad; 1:5,000). Chemiluminescence images were acquired on a Chemidoc Touch imaging system (Bio-Rad) using Clarity western ECL substrate (Bio-Rad).

# Supplementary Information

# Acknowledgements

SD Liston was supported by a Postdoctoral Fellowship from the Natural Sciences and Engineering Research Council of Canada (516840-2018). Crystal and cryo-EM structures solved in this work were funded in whole or in part with US federal funds from the National Institute of Allergy and Infectious Diseases, National Institutes of Health, Department of Health and Human Services, under contract No. HHSN272201700060C (Center for Structural Genomics of Infectious Diseases (CSGID); http://csgid.org). LE Cowen is funded by a Canadian Institutes of Health Research (CIHR) Foundation grant (FDN-154288) and is a Canada Research Chair (Tier 1) in Microbial Genomics & Infectious Disease and co-director of the CIFAR Fungal Kingdom: Threats & Opportunities program. We thank Merck and Genome Canada for making the GRACE *C. albicans* mutant collection available and Andre Nantel and the National Research Council of Canada for their distribution. We thank all members of Savchenko and Cowen laboratories for their help and assistance.

## Author Contributions

PJ Stogios: conceptualization, data curation, formal analysis, validation, investigation, visualization, methodology, and writing—original draft, review, and editing.
SD Liston: data curation, validation, methodology, and writing—original draft.
C Semper: conceptualization, data curation, formal analysis, validation, methodology, and writing—original draft.
B Quade: data curation, validation, methodology, and writing—original draft.
K Michalska: data curation and validation.
E Evdokimova: data curation.
S Ram: data curation.
Z Otwinowski: validation and methodology.
D Borek: data curation, methodology, and writing—original draft.
LE Cowen: conceptualization, supervision, funding acquisition, methodology, and writing—original draft.
A Savchenko: conceptualization, data curation, formal analysis, supervision, funding acquisition, validation, investigation, methodology, project administration, and writing—original draft, review, and editing.

## Conflict of Interest Statement

LE Cowen is a co-founder and shareholder in Bright Angel Therapeutics, a platform company for development of novel antifungal therapeutics. LE Cowen is a consultant for Boragen, a small-molecule development company focused on leveraging the unique chemical properties of boron chemistry for crop protection and animal health. SD Liston was supported by a Mitacs postdoctoral fellowship funded in partnership with Amplyx Pharmaceuticals Inc., now a subsidiary of Pfizer. D Borek and Z Otwinowski are co-founders of Ligo Analytics, a company that develops a software for cryogenic electron microscopy. Z Otwinowski is a co-founder of HKL Research, a company that develops software for X-ray diffraction data analysis.

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
