## [Reviewer comments · Life Science Alliance]

Life Science Alliance

Molecular analysis and essentiality of Aro1 shikimate biosynthesis multi-enzyme in *Candida albicans*

Alexei Savchenko, Peter Stogios, Sean Liston, Cameron Semper, Bradley Quade, Karolina Michalska, Elena Evdokimova, Shane Ram, Zbyszek Otwinowski, Dominika Borek, and Leah Cowen

DOI: <https://doi.org/10.26508/lsa.202101358>

Corresponding author(s): Alexei Savchenko, University of Calgary

Review Timeline:	Submission Date:	2021-12-30
	Editorial Decision:	2022-02-16
	Revision Received:	2022-03-27
	Editorial Decision:	2022-04-01
	Revision Received:	2022-04-10
	Accepted:	2022-04-13

Scientific Editor: Novella Guidi

Transaction Report:

February 16, 2022

Re: Life Science Alliance manuscript #LSA-2021-01358-T

Dr. Alexei Savchenko
University of Calgary
Microbiology, Immunology and Infectious Diseases
3330 Hospital Drive NW
3330 Hospital Drive NW
Calgary, Alberta T2N 4N1
Canada

Dear Dr. Savchenko,

Thank you for submitting your manuscript entitled "Molecular analysis and essentiality of the Aro1 shikimate biosynthesis multi-enzyme in *Candida albicans*" to Life Science Alliance. The manuscript was assessed by expert reviewers, whose comments are appended to this letter. We, thus, encourage you to submit a revised version of the manuscript back to LSA that responds to all of the reviewers' points.

Thank you for this interesting contribution to Life Science Alliance. We are looking forward to receiving your revised manuscript.

Sincerely,

B. MANUSCRIPT ORGANIZATION AND FORMATTING:

Reviewer #1 (Comments to the Authors (Required)):

In this paper, Stogios et al present a structural and functional analysis of *C. albicans* Aro1, a key enzyme in the shikimate pathway, which carries 5 independent enzymatic activities. Since this type of enzymes is fungal-specific and essential, it represents a potential target for antifungal drugs, by analogy with glyphosate, an herbicide that targets the plant EPSP synthase. The Aro1 structure of a non-pathogenic fungus, *Chaetomium thermophilum* AroM, was previously published. Here, the authors show that the *C. albicans* Aro1 structure is very similar to that of AroM. They analyze the effect of mutations in the five catalytic sites on activity in vitro and in vivo, and find that several of them are essential for cell viability. Surprisingly, they find that one of the catalytic domains, DHQase, is inactive. This activity is carried out by a distinct enzyme, type II DHQase, which is found in several species that have a mutated Aro1 DHQase, but absent in species having active Aro1 DHQase. Overall, this detailed analysis represents a sound basis for the future identification of Aro1 inhibitors as new antifungal drugs.

I am not competent to comment on the structural data acquisition and analysis. The functional analyses are overall convincing. The writing is clear, but as a general comment, figure legends could be more detailed.

Specific remarks:

(Note - in the absence of line numbering or even page numbering, I copied the commented sentences).

Table 1: Aro1 mutant H268K/H280K was expressed in *E. coli*, whereas the wild-type protein could not be expressed in *E. coli* but was expressed in *C. albicans* instead. One concern is that activity of the enzyme expressed in different organisms will be different based on modifications, metal cofactor incorporation etc. The very fact that the wild-type enzyme could only be expressed in *C. albicans* vs. the mutant enzyme also in *E. coli*, points to the possibility of such differences. Why could not all mutants be purified from *C. albicans* like the wild-type Aro1?

Fig. 4A vs Fig S9: many mutants show a much stronger phenotype by colony formation (Fig. 4A) vs growth in liquid medium (Fig. S9): 1334, 1370, DHSDel, possibly also 890, 980. These discrepancies should be discussed.

- "Similarly to *C. albicans* tetO-ARO1/tetO-ARO1, *C. albicans* tetO-His6TEVARO1/tetO-His6TEVARO1 had a growth defect on solid (Fig. 4a) and in liquid medium (Supplementary Fig. 9).": this, presumably, should read "... in medium containing doxycycline".

- Fig. 4, legend vs text: the text refers to the construct tetO-His6TEVARO1, whereas the legend on the figure has tetO-His6ARO1. This should be harmonized for clarity.

- Fig S8: the legend on the figure marks the constructs tetO-His6TEVARO1 and pACT1-His6ARO1. I presume that the TEV site is present in both constructs, so it should be indicated.

- End of discussion, "We thus conclude that the presence of a more efficient type II DHQase in *C. albicans* may have alleviated selective pressure for maintaining DHQase activity within Aro1. Similarly, the presence of intact DHQase domains in the Aro1 enzymes of *C. glabrata* and *C. parapsilosis* obviated evolution of type II DHQase in these species.": First, the authors suggest that type II DHQase specifically evolved in the species having lost DHQase activity of Aro1 ("the presence ... obviated evolution of type II DHQase in these species"), rather than being maintained in these species. They should present evidence for this suggestion. Second, rather than "similarly", the relation between the two sentences appears to me to be "conversely" because they present two alternative hypotheses: either loss of the DHQase activity of Aro1 in *C. albicans* et al. led to maintenance of type II DHQase activity, or loss of type II DHQase in *C. parapsilosis* et al. led to maintenance of the DHQase activity of Aro1.

Typos

Supplementary Fig. 5: panel C is wrongly marked B

Legend of Fig S8, a) AND c): "Cultures were grown in SC medium overnight at 30 °C, subcultured into SC medium

supplemented with 0.050 µg/mL DOX overnight, then subcultured into SC medium supplemented with 5 µg/mL DOX for 4 h."

Reviewer #2 (Comments to the Authors (Required)):

Fungi are capable of synthesizing aromatic amino acids de novo, starting from phosphoenolpyruvate and erythrose-4-phosphate that are converted to chorismate. Five consecutive reactions of this so-called shikimate pathway are carried out by a penta-functional multi-domain enzyme. About two years ago, another group has reported on the structure of this enzyme from *C. thermophilum*. Here, the authors describe the structural and biochemical investigation of the same multi-functional protein from the *C. albicans*, a human pathogen. The amount of work presented in this manuscript is impressive and the authors must be congratulated for such a piece of work. The manuscript does indeed describe the crystal structures of three enzyme fragments, the cryoEM structure of the entire system, the elucidation of functionally critical residues, and the discovery that one domain of the protein is catalytically dead with its function being taken over by another protein. I am convinced that this work adds considerably to our understanding of the structural enzymology and evolution of the shikimate pathway. The authors should, however, improve the clarity of the presentation of their beautiful work.

- Figure 1 should show the reactions in a chemdraw style to help the reader to follow such a complex story.
- The description of the full-length structure should be improved. A first point concerns the cryoEM map interpretation. Can the authors trace the entire protein chain in the map? With a considerable degree of flexibility, how was it possible to assign each domain to a subunit rather than the other within the dimer? Is domain swapping possible between the two subunits?
- The pictures should outline where the active sites are located.
- Does the active site proximity somehow match the order of the reactions in the metabolic pathway. I honestly found the analysis of possible channeling effects unsatisfactory. What about the position and orientation of substrate-binding sites and their openings in the complex? Is it possible that they exchange their substrate/products?
- Tables S1 and S2 are incomplete. Something went wrong in their preparation. The CC1/2 values for two structures are not listed. The cells supposed to list the refinement statistics of the cryoEM structures are empty. Both tables contain several formatting errors.
- Why could the H268K/H280K double mutant be expressed in *E. coli* whereas the WT protein did not work out? Any hypothesis? Is the protein toxic to *E. coli*?

Referee Cross-Comments

Both reviewers reached similar conclusions. Nothing to add.

We would like to thank the editor of Life Science Alliance and the two reviewers for their constructive comments and suggestions. We have carefully reviewed all of them and adjusted the manuscript to address them in full. Below we provide the specific clarifications and comments for each point in reviewers' letters.

Given overall positive response our manuscript received from the editor and reviewers we are convinced that changes and adjustments made based on reviewers comments have improved our draft to merit the publication of this original research data in Life Science Alliance.

Replies to the specific the specific reviewers

Reviewer #1:

-Table 1: Aro1 mutant H268K/H280K was expressed in E. coli, whereas the wild-type protein could not be expressed in E. coli but was expressed in C. albicans instead. One concern is that activity of the enzyme expressed in different organisms will be different based on modifications, metal cofactor incorporation etc. The very fact that the wild-type enzyme could only be expressed in C. albicans vs. the mutant enzyme also in E. coli, points to the possibility of such differences. Why could not all mutants be purified from C. albicans like the wild-type Aro1?

In response to reviewer's question about expressing and purifying the Aro1 mutants from *C. albicans* we would like to point out that this is not possible due to the essentiality of all four of the catalytically active domains within Aro1. Because of the essential nature of these enzyme activities, mutation of both Aro1 alleles in *C. albicans* would result in a lethal phenotype. The dimeric nature of the protein would result in a mixed population of WT-Aro1/mutant-Aro1, even if only the mutant allele had been 6His-tagged in a heterozygous mutant.

To ensure an accurate interpretation of the consequences of these active site mutations, we were therefore required to express them recombinantly in *E. coli*. Regarding concern over differential levels of enzymatic activity in Aro1 protein purified from *C. albicans* vs. *E. coli*, we have included an additional supplemental Figure S9 that clearly shows the comparable levels of activity in the EPSP synthase, shikimate kinase, and shikimate dehydrogenase between native (from *C. albicans*) and recombinantly expressed (from *E. coli*) Aro1 proteins. Please note that we are unable to compare activity of the 3-dehydroquinate synthase domain because we were not

able to expression the full-length Aro1 with this domain intact in *E. coli*, likely due to instability of this construct.

To clarify this point in the text of the manuscript we added the following sentence to the Results section:

Catalytic activity in other Aro1 domains was unaffected by this mutation (Supplementary Figure 9).

And we also added the caption for the new Supplementary Figure 9:

Figure S9. Comparison of enzymatic activity between Aro1 purified from *C. albicans* and recombinant Aro1 purified from *E. coli*

Enzymatic activity, expressed as percent relative to wild type, is shown. The data depicted for the EPSPS, SK, and DHSD domains is from recombinant full-length enzyme purified from *E. coli*. The wild-type data is from *C. albicans* Aro1 and normalized to 100%.

Addition of Supplementary Fig. 9 necessitated the changing of the order in the original Supplementary Figures 9, 10 and 11 which are now Supplementary Figures 10, 11 and 12.

Fig. 4A vs Fig S9: many mutants show a much stronger phenotype by colony formation (Fig. 4A) vs growth in liquid medium (Fig. S9): 1334, 1370, DHSDdel, possibly also 890, 980. These discrepancies should be discussed.

Observed differences in severity of growth defects on solid versus liquid media are quite common and usually originate from the difference in the starting inoculum using for each of these complementary assays or differences in growth phenotypes in the different conditions. In liquid assays, overnight cultures in 96-well plates were pinned (~1uL inoculum) into new plates containing 80uL of fresh medium for OD readings over time. Whereas the spotting assays were performed with 10-fold dilution series of cells. Another potential contributor may be that growth defects triggered by mutations in *ARO1* result in dramatic change of cells size and cell death which we observed and reported in our manuscript (see Supplementary Fig. 11). These conditions would affect the OD reading that is used for the liquid assay. Specifically, the release of the DNA by dying cells would cause cell clumping, which could interfere with OD readings and, as a result, with direct comparison to wild-type cell culture. However, since the main purpose of both assays was to establish the effect of specific Aro1 domain inactivation on viability, the results of both assays are coherent and point to essentiality of specific Aro1 activities.

As per reviewer's suggestion we added a sentence explaining this phenomenon in the discussion section of the manuscript (the added sentence is highlighted in yellow):

Strong growth defects and cell death were also observed in case of *C. albicans* strains carrying active site mutations in the SK and DHSD domains. The difference in severity of observed effects between the growth on solid and liquid media may be due to significant clumping of the cells due to cell lysis that would interfere with optical density reading used for the latter assay. These data suggest that *C. albicans* lacks enzymes that could provide compensatory activities to support growth in the absence of the DHQS, EPSPS, SK, and DHSD activities of Aro1, at least under the cultivation conditions tested in this study.

- "Similarly to *C. albicans tetO-ARO1/tetO-ARO1*, *C. albicans tetO-His6TEVARO1/tetO-His6TEVARO1* had a growth defect on solid (Fig. 4a) and in liquid medium (Supplementary Fig. 9).": this, presumably, should read "... in medium containing doxycycline".

According to reviewer's suggestion we added: "and in liquid medium (Supplementary Fig. 10) when doxycycline was added."

- Fig. 4, legend vs text: the text refers to the construct *tetO-His6TEVARO1*, whereas the legend on the figure has *tetO-His6ARO1*. This should be harmonized for clarity.

We thank the reviewer for noticing this discrepancy. Figure 4 has been amended to correct this error.

- Fig S8: the legend on the figure marks the constructs *tetO-His6TEVARO1* and *pACT1-His6ARO1*. I presume that the TEV site is present in both constructs, so it should be indicated.

In response to this comment, we would like to clarify that the Figure S8 is labelled correctly since this strain is *tetO-Aro1/pACT1-His6TEVAro1* (See Materials and Methods regarding construction of strain CaLC6598).

-End of discussion, "We thus conclude that the presence of a more efficient type II DHQase in *C. albicans* may have alleviated selective pressure for maintaining DHQase activity within Aro1. Similarly, the presence of intact DHQase domains in the Aro1 enzymes of *C. glabrata* and *C. parapsilosis* obviated evolution of type II DHQase in these species.": First, the authors suggest that type II DHQase specifically evolved in the species having lost DHQase activity of Aro1 ("the presence ... obviated evolution of type II DHQase in these species"), rather than being maintained in these species. They should present evidence for this suggestion. Second, rather than "similarly", the relation between the two sentences appears to me to be "conversely" because they present two alternative hypotheses: either loss of the DHQase activity of Aro1 in *C. albicans et al.* led to maintenance of type II DHQase activity, or loss of type II DHQase in *C. parapsilosis et al.* led to maintenance of the DHQase activity of Aro1.

The reviewer brings up an important point that these two hypotheses are discordant. We have addressed this comment by changing this portion of the manuscript to read:

“We hypothesize that the presence of a more efficient type II DHQase in *C. albicans* may have alleviated selective pressure for maintaining DHQase activity within Aro1. Similarly, the lack or loss of a type II DHQase ortholog in *C. glabrata* and *C. parapsilosis* necessitated maintenance of the catalytic dyad and consequently, enzymatic activity, in the DHQase domain of Aro1 in these species. “

Reviewer #2:

-Figure 1 should show the reactions in a chemdraw style to help the reader to follow such a complex story.

We agree with this suggestion. We improved Figure 1 according to this suggestion, which now shows the chemical structure of the major compounds in the shikimate pathway. This necessitated increasing the overall size of the Figure 1.

-The description of the full-length structure should be improved. A first point concerns the cryoEM map interpretation. Can the authors trace the entire protein chain in the map? With a considerable degree of flexibility, how was it possible to assign each domain to a subunit rather than the other within the dimer? Is domain swapping possible between the two subunits?

We modified the description of the full-length structure in the text of the manuscript and updated tables.

Briefly, the cryoEM analysis is based on three structural maps:

- 4.19 Å map that corresponds to the homogeneous refinement based on 3D reconstruction of full length Aro1 dimer (EMDB: 26357, PDB: 7U5S). In this map we performed a rigid body fit of two high-resolution atomic models of Aro1_{DHQs-EPSPs} and Aro1_{SK-DHQase-DHSD} obtained as described below.
- 3.43 Å map that corresponds to the local refinement based on focused 3D reconstruction of Aro1_{DHQs-EPSPs} dimer (EMDB: 26358, PDB: 7U5T). We performed atomic refinement of X-ray crystallographic models with PHENIX for this map and used a monomer consisting of two domains (DHQS-EPSPS) for fitting to the 4.19 Å map describing the full-length structure.
- 3.16 Å map that corresponds to the local refinement based on focused 3D reconstruction of Aro1_{SK-DHQase-DHSD} (EMDB: 26359, PDB: 7U5U). We performed atomic refinement of X-ray crystallographic models with PHENIX for this map and used a monomer consisting of three domains (SK-DHQase-DHSD) for the fitting to 4.19 Å map describing the full-length structure.

Can the authors trace the entire protein chain in the map?

Indeed, obtained density map allow us to trace the entire protein chain. However, we did not trace or build models independently for the full-length structure. Instead, we used X-ray crystallographic models obtained based on high-resolution diffraction data, performed fitting and atomic refinement with these models to two higher-resolution cryoEM maps obtained with focused 3D reconstruction, and then placed refined entities (Aro1_{DHQs-EPSPS} and Aro1_{SK-DHQase-DHSD}) using real space fitting functions available in Chimera in combination with rigid body refinement with PHENIX to obtain the model of the full-length structure. Consequently, the full-length structure model is missing only the linker between Aro1_{DHQs-EPSPS} and Aro1_{SK-DHQase-DHSD} domains, and a few very short surface loops that were absent in cryoEM and X-ray atomic models refined to higher resolution. The process of real-space fitting resembles molecular replacement in X-ray crystallography. The rigid body refinement resulted in good agreement with the map and good agreement of main chain and large side chains with the 4.19 Å map.

With a considerable degree of flexibility, how was it possible to assign each domain to a subunit rather the other within the dimer?

The flexibility in the full-length dimeric structure arises from changes in orientation between Aro1_{DHQs-EPSPS} and Aro1_{SK-DHQase-DHSD} for each monomer, but each group of domains behaves as a rigid body—this is one of the reasons why we could reconstruct these groups of domains to higher resolution with focused classification.

Aro1_{DHQs-EPSPS} and Aro1_{SK-DHQase-DHSD} rotate in respect to each other for each Aro1 monomer in dimeric structure, however their relative orientations are restricted because they are connected by a short linker (10 to 11 aa). We were not able to confidently trace this linker but density for it is visible in the map contoured at lower level (this data is added as a new panel in Supplementary Figure 5c) and suggests no domain swapping. In addition, even fully extended unmodeled linker of 10-11 aa (with ~3.25 Å per aa in fully extended conformation would amount to ~35 Å in length) is not sufficient to allow for domain swapping between domains of chain A and chain B as the two ends (residue 846 and residue 857 in each group domains) are separated by ~100 Å between Aro1_{DHQs-EPSPS} (chain A-Pro846) and Aro1_{SK-DHQase-DHSD} (chain B-Ser857) and by ~80 Å between Aro1_{DHQs-EPSPS} (chain B-Pro846) and Aro1_{SK-DHQase-DHSD} (chain A-Ser857). These calculations suggest degree of asymmetry between two orientations rather than domain swapping. We updated the main text in Results section accordingly and updated Supplementary Figure 5c to show the linker contoured at lower level (changes are highlighted in yellow):

“The 4.19 Å map that corresponds to the homogeneous refinement based on 3D reconstruction of full length Aro1 dimer (EMDB: 26357, PDB: 7U5S). In this map we performed only a rigid body fit of two high-resolution models for Aro1_{DHQs-EPSPS} and Aro1_{SK-DHQase-DHSD} domains. These two models (Aro1_{DHQs-EPSPS} and Aro1_{SK-DHQase-DHSD}) were obtained by docking and refining against cryo-EM maps X-ray crystallographic models described earlier. We performed atomic refinement of X-ray crystallographic models with PHENIX and corrected structures manually

with Coot. The molecular maps at lower contour level indicated where the linker between Aro1_{DHQS-EPSPS} and Aro1_{SK-DHQase-DHSD} is located (Supplementary Fig. 5c). Analysis of the possible distances for 11 aa unmodeled linker excluded the possibility of domain swapping.”

The domains that group together in the Aro1 structure i.e. Aro1_{DHQS-EPSPS} and Aro1_{SK-DHQase-DHSD} fit unambiguously in the cryoEM derived map of the full-length protein. The higher resolution cryoEM maps can be considered submaps of the map corresponding to the full-length protein, so the relative domain orientations derived from each of these maps is preserved in the full-length structure.

-Why didn't we perform atomic resolution refinement for the full-length reconstruction?

The resolution 4.19 Å is overall resolution corresponding to 0.143 CTF criterion, but resolution varies across the map and in some areas is higher than the others. Therefore, although we are confident in overall fit of each domain, we were mindful of potential mis-tracings, particularly in case of beta sheets, which typically require resolution better than 3.9 Å for clear separation of beta strands or in loops which tend to be more flexible. Finally, we decided that atomic refinement would mislead the readers. The key information provided by cryoEM data was orientation of individual domains with respect to each other in context of the full-length Aro1 and the 4.19 Å map, with fitting results of focused classification obtained to higher resolutions, provides this information. However, information about positions of each atom in the structure should be derived from higher resolution structures obtained either from local refinement after focused classification or from X-ray crystallographic data that is provided in our manuscript.

-The pictures should outline where the active sites are located.

We agree with this suggestion and modified Figure 3B to show the locations of the active sites of the Aro1 domains. We also modified Figure 3B to label the two subunits in the dimer.

-Does the active site proximity somehow match the order the reactions in the metabolic pathway. I honestly found the analysis of possible channeling effects unsatisfactory. What about the position and orientation of substrate-binding sites and their openings in the complex? Is it possible that they exchange their substrate/products?

The reviewer brings up some excellent questions that we have also thought about and agree that commenting on these questions would improve the manuscript. When visualized in the cryo-EM SPR 3D dimeric structure of Aro1, the domains are oriented spatially in the order of the reaction in a “counter-clockwise” arrangement from “bottom-right” to “bottom-left”. However, the active sites all face outward away from the dimerization interface and do not co-localize with each other. There are also no obvious clefts/tunnels or structures on the dimer that would suggest a path for movement of substrates/products between the active sites. These observations imply

that there are no exchange or channeling of compounds between active sites, however, further study outside of the main scope of this work would be required to address this possibility.

We updated the Results and Discussion sections accordingly to discuss these observations and implications (changes are highlighted in yellow).

Revisions to Results section:

The analysis of the full-length Aro1 structure highlighted the high level of structural similarity between the specific domains in the Aro1 dimer and their individual crystal structures (Supplementary Fig. 6). The Aro1_{DHQs} domain, Aro1_{EPSPS} subdomain II, and Aro1_{SK-DHQase-DHSD} backbones show nearly identical conformation within the core structure of each region with minor differences seen in some loop regions. This implies that the large movement of the Aro1_{SK-DHQase-DHSD} region does not drive structural changes that modulate substrate binding or catalysis, but instead reorients and repositions the domains. When visualized in the molecular structure, the positioning of the domains of Aro1 matches the order of reactions in the pathway in a counter-clockwise fashion, with the domain that catalyzes the first reaction (Aro1_{DHQs}) positioned at the “bottom” of the dimer, with the next two domains (Aro1_{DHQase} and Aro1_{DHSD}) positioned sequentially above Aro1_{DHQs}, followed by Aro1_{SK} at the “top left” and terminating with Aro1_{EPSPS} at the “bottom left” (Fig. 3C). However, each individual active site is more than 40 Å from each other and each face outwards from the core of the dimer (Fig. 3C). We also do not observe any obvious clefts or tunnels between active sites.

Revisions to Discussion:

The impact of assembly of multiple enzymatic domains into a single Aro1/AroM protein for their catalytic activity remains elusive. *In vitro* characterization of *C. thermophilum* AroM concluded that concatenation of domains catalyzing consecutive enzymatic reactions into a single protein does not increase the flux through the pathway (Arora Veraszto et al., 2020). As in *C. thermophilum* AroM, our cryo-EM SPR structure of *C. albicans* Aro1 did not reveal that the individual active sites are spatially co-localized, nor is there evidence of any structural features connecting them. These observations suggest that there is no exchange of substrates/products between active sites and concatenation of the domains in *C. albicans* Aro1 should not increase the flux through the pathway, however this remains to be experimentally deduced. Given our observation that the DHQase domain in this multienzyme is inactive, there are clearly differences between fungal Aro1/AroM that remain to be explored. The relative orientation between the Aro1_{SK} and Aro1_{EPSPS} domains changes dramatically between “tight” and “loose” conformations of Aro1 protomers (Fig. 3), which may facilitate diffusion of substrates between these domains and support flux between consecutive steps supported by these Aro1 domains. The “blurring” of the subdomain I of Aro1_{EPSPS} in our cryoEM SPR analysis suggests local movement of this

fragment. The exchange from “loose” to “tight” conformation may restrict the movement of this portion of Aro1_{EPSPS} and promote the catalytically active positioning between the two Aro1_{EPSPS} subdomains.

-Tables S1 and S2 are incomplete. Something went wrong in their preparation. The CC1/2 values for two structures are not listed. The cells supposed to list the refinement statistics of the cryoEM structures are empty. Both tables contain several formatting errors.

We thank the reviewer for noting this, the formatting error must have arisen from copying between software packages.

For Table S1, we were able to fill in the CC1/2 values for the highest resolution shells as this was neglected in our original submitted paper. The legend has been updated to indicate that these values correspond to the highest resolution shells. The overall CC1/2 values were unavailable as the version of HKL3000 we used to process the data was an older version that did not output these values. However, the overall quality of data sets was excellent as shown by other statistical indicators of quality. In particular, the values of CC1/2 for the highest resolution shells indicate that we didn't overestimate the resolution of the diffraction patterns.

Table S2 has been completed.

The formatting errors have been fixed in both tables.

General edits:

In addition to the edits proposed by the reviewers we have also fixed following sentences in the main text, which had the typos that were missed in the original manuscript.

*Updated the sentence to fix typos:

The regions corresponding to residues 844 to 857 and 967 to 984 were not modeled due to poor electron density.

To:

The regions corresponding to residues 844 to 855 and 970 to 984 were not modeled due to poor electron density.

*Updated the sentence as it was incomplete:

The latter region corresponds to the 'lid loop' or 'P-loop' that is involved in positioning of shikimate in the active site (Gu et al., 2002, Hartmann et al., 2006, Sutton et al., 2015).

To:

The latter region corresponds to the 'lid loop' or 'P-loop' that is involved in positioning of **ATP** **and** shikimate in the active site (Gu et al., 2002, Hartmann et al., 2006, Sutton et al., 2015).

April 1, 2022

RE: Life Science Alliance Manuscript #LSA-2021-01358-TR

Dr. Alexei Savchenko
Univesty of Calgary
Microbiology, Immunology and Infectious Dlseases
3330 Hospital Drive NW
3330 Hospital Drive NW
Calgary, Alberta T2N 4N1
Canada

Dear Dr. Savchenko,

Thank you for submitting your revised manuscript entitled "Molecular analysis and essentiality of Aro1 shikimate biosynthesis multi-enzyme in *Candida albicans*". We would be happy to publish your paper in Life Science Alliance pending final revisions necessary to meet our formatting guidelines.

- please correct some typos that were pointed out by Reviewer 1, in the legend of Fig. S8 a and c
- please add scale bars for figures 3A and S5A
- please include the supplementary references section into the main references section
- please upload tables in separate files
- please add Keywords and a Category to our manuscript system
- please add the Twitter handle of your host institute/organization as well as your own or/and one of the authors in our system
- please use the [10 author names, et al.] format in your references (i.e. limit the author names to the first 10)
- please double-check your figure legends for Figure S12 and label the panels correctly
- please add a callout for Figure S5B to your main manuscript text

A. FINAL FILES:

B. MANUSCRIPT ORGANIZATION AND FORMATTING:

Sincerely,

Reviewer #1 (Comments to the Authors (Required)):

The authors have satisfactorily addressed the main criticisms raised in the previous round of reviewing. I notice however that they failed to correct some typos that were pointed out, in the legend of Fig. S8 a and c(subculuted). Otherwise I have no more concerns.

Reviewer #2 (Comments to the Authors (Required)):

I confirm my evaluation: this is very good and relevant work. The authors have addressed both reviewers' comments in a convincing and satisfactory way.

April 13, 2022

RE: Life Science Alliance Manuscript #LSA-2021-01358-TRR

Dr. Alexei Savchenko
University of Calgary
Microbiology, Immunology and Infectious Diseases
3330 Hospital Drive NW
Calgary, Alberta T2N 4N1
Canada

Dear Dr. Savchenko,

Thank you for submitting your Research Article entitled "Molecular analysis and essentiality of Aro1 shikimate biosynthesis multi-enzyme in *Candida albicans*". It is a pleasure to let you know that your manuscript is now accepted for publication in Life Science Alliance. Congratulations on this interesting work.

DISTRIBUTION OF MATERIALS:

Again, congratulations on a very nice paper. I hope you found the review process to be constructive and are pleased with how the manuscript was handled editorially. We look forward to future exciting submissions from your lab.

Sincerely,
